# The Estimated Intake of S100B Relates to Microbiota Biodiversity in Different Diets

**DOI:** 10.3390/biom15071047

**Published:** 2025-07-18

**Authors:** Tehreema Ghaffar, Veronica Volpini, Serena Platania, Olga Vassioukovitch, Alessandra Valle, Federica Valeriani, Fabrizio Michetti, Vincenzo Romano Spica

**Affiliations:** 1Laboratory of Epidemiology and Biotechnologies, University of Rome “Foro Italico”, 00135 Rome, Italy; t.ghaffar@studenti.uniroma4.it (T.G.); v.volpini@studenti.uniroma4.it (V.V.); alessandravalle2000@gmail.com (A.V.); federica.valeriani@uniroma4.it (F.V.); 2Genes Research Start Up, 00187 Rome, Italy; platania@genes4you.it (S.P.); fabriziomichetti.office@gmail.com (F.M.)

**Keywords:** S100B, gut microbiota, diet, Shannon index, food, chronic disease prevention, dietary patterns

## Abstract

The S100B protein, known for its role in the central and enteric nervous systems, has recently been identified in dietary sources such as milk, dairy products, fruits, and vegetables. Given its potential interaction with the gut microbiota, this study explores the relationship between dietary intake of S100B and microbiota biodiversity across different diets. A comprehensive study was conducted, estimating S100B concentrations in 13 dietary patterns recommended in different countries. This is the first study to provide a comparative estimation of S100B exposure from the diet and to explore its potential ecological and epidemiological relevance. The association between S100B levels and microbiota biodiversity was statistically analyzed, showing a direct correlation. Microbial diversity was assessed using the Shannon index, based on data extracted from studies reporting microbiota composition across dietary patterns. Additionally, the relative risk of Crohn’s disease was assessed in different populations to examine potential links between dietary patterns, S100B, and chronic disease prevention. A moderate positive correlation (R^2^ = 0.537) was found between S100B concentration and Shannon index, suggesting that diets higher in S100B (e.g., Mediterranean diet) were associated with higher microbial alpha-diversity. Furthermore, Western-style diets, with the lowest S100B levels, exhibited a higher relative risk for Crohn’s disease (R^2^ = 0.780). These findings highlight the potential role of dietary S100B content in modulating gut microbiota diversity and reducing chronic disease risk.

## 1. Introduction

Recent data have allowed a change in perspectives concerning the role(s) of S100B protein, showing the presence of its molecular motif in diet [1]. S100B is a calcium-binding protein that is concentrated in astrocytes within the central nervous system (CNS), while also being expressed in other neural and non-neural cell types, including enteroglial cells in the enteric nervous system (ENS) [2]. This macromolecule is regarded as a member of the S100 protein family, which at present includes more than 20 genes. All S100 proteins are characterized by the presence of two Ca^2+^-regulated motifs, which are linked by a connecting hinge region, thus resulting in a helix–loop–helix (HLH) structure (Figure 1) [3]. The S100B protein was the first to be discovered among this gene family, and was originally regarded as being specific to the nervous system. Interestingly, it is highly conserved during phylogeny, suggesting a key role in different species. The biological effect(s) of the protein appears to be concentration-dependent, i.e., trophic at low nanomolar physiological concentrations and toxic at high micromolar pathologic concentrations, and its levels in biological fluids represent a marker of active injury in neural and even extra-neural tissues [1,4]. Beyond its endogenous roles, S100B has recently been detected in commonly consumed foods, including milk, dairy products, vegetables, and common fruits such as apples [5]. While the protein has been extensively studied in animals, in silico analyses have also predicted the presence of S100B-like immunoreactive sequences in various edible plant species [5]. Although structurally similar, the biological activity of these plant-derived motifs remains unclear, and it is still unknown whether they retain the functional properties of animal-derived S100B. The potential dietary origin of S100B opens new perspectives on its interaction with the human body, particularly through the gut.

In this way, the protein might interact with the entire body owing to its recently demonstrated interaction with gut microbiota, which is known to regulate a series of functions in different organs, including nervous activities through the gut–brain axis [6,7]. This interaction is supported by the presence of S100B in enteric glial cells, which are known to play a key role in microbiota regulation and gut homeostasis. Furthermore, recent in vivo experiments on murine models have shown that oral S100B supplementation modulates gut microbial diversity [5]. These findings suggest that S100B may act as a trophic or signaling molecule within the gut–brain axis, influencing microbial ecology and potentially contributing to health maintenance.

Recent studies suggest that a lower diversity in the gut microbiota is linked to intestinal inflammation and Crohn’s disease. S100B, which is overexpressed in inflamed gut tissue, may influence these processes by affecting immune responses and gut barrier function through the RAGE pathway. Therefore, differences in dietary S100B could potentially impact gut health and inflammation [4]. This study specifically aims to evaluate the estimated dietary intake of S100B across 13 global dietary patterns and to examine its association with microbiota alpha-diversity (Shannon index) and the relative risk of chronic diseases, including Crohn’s disease. To our knowledge, this is the first study to estimate potential S100B exposure through diet and explore its ecological and epidemiological significance at the population level. The results suggest a correlation between S100B content and microbiota biodiversity as an indirect and promising index of good health.

## 2. Materials and Methods

### 2.1. Search Strategy

A comprehensive literature search was conducted in PubMed, Scopus, and Web of Science for studies published between 2000 and 2024 (Appendix A). The search included keywords (“dietary patterns” OR “healthy diet” OR “nutritional patterns” OR “food intake”) AND (“meta-analysis” OR “systematic review” OR “cohort study” OR “randomized controlled trial”) AND (2000:2024[Date-Publication]).

### 2.2. Inclusion/Exclusion Criteria

To ensure high-quality evidence, only randomized controlled trials (RCTs) and observational cohort studies evaluating the association between dietary patterns and chronic disease risk were included. Studies were considered eligible if they had a follow-up period of at least five years, providing a long-term perspective on the effects of dietary interventions. Moreover, only studies that reported relative risk (RR) or hazard ratios (HR) with confidence intervals (CI) were selected, ensuring consistency in the measurement of outcomes. Importantly, the studies had to assess dietary patterns holistically, rather than focusing solely on individual nutrients, allowing for a more comprehensive understanding of the effects of whole diets on chronic disease prevention. Studies were excluded if they lacked sufficient data or did not employ a clear methodology for dietary assessment, as these factors could introduce bias or limit the interpretability of results. Reports conducted exclusively on animal models or in vitro experiments were omitted, as the findings may not be directly translatable to human populations. Additionally, editorials, commentaries, and opinion pieces were excluded from the meta-analysis to avoid redundancy and ensure that the focus remained on primary research. Finally, studies with a high risk of bias, as determined by the Cochrane Risk of Bias Tool, were excluded to maintain the integrity of the findings.

### 2.3. Data Extraction and Quality Assessment

Data were extracted independently by two reviewers and cross-checked for accuracy. Extracted information included study design, population characteristics, dietary patterns analyzed, chronic diseases investigated, and risk estimates. The Newcastle–Ottawa Scale (NOS) was used to assess the quality of observational studies, while the Cochrane Risk of Bias Tool was applied to RCTs. Disagreements were resolved by discussion or consultation with a third reviewer.

### 2.4. Estimation of S100B in Different Dietary Patterns

To calculate the S100B concentrations in different dietary regimens, a comprehensive list of 13 dietary patterns recommended in different countries was compiled based on evidence from the scientific literature and national dietary guidelines (Appendix A). These patterns reflect culturally and regionally diverse diets that are recurrently reported in public health studies and endorsed in national nutrition policies. The serving sizes of dietary components were accessed using the validated food guide pyramids of each region and country as officially released by their health ministries or national associations. The main components of interest for the calculation of S100B were milk (from cow and/or sheep and/or goat), dairy products (e.g., ricotta, yogurt, kefir, cheese), fruits (e.g., apples, durian, jack fruit), and vegetables (e.g., spinach, Brassicaceae). The different species and the total values of S100B concentration in each diet were estimated using the reference values of S100B for dietary components by following the reference values provided by Michetti et al., 2025 [1]. Given the wide variability in S100B concentrations reported in different milk sources (e.g., cow, goat, sheep), dairy products, and other foods, we opted to use mean values as a conservative and reproducible metric to approximate average dietary exposure across populations. Moreover, a quantitative enzyme-linked immunosorbent assay (ELISA) for detecting S100B protein (Abcam ab234573, GR3360381-1, Cambridge, UK, Millipore Merck EZHS100B-33K, Billerica, MA, USA and MyBioSource, MBS2503148, San Diego, CA 92195-3308, USA) was performed on samples from different dairy and plants sources to better estimate the S100B concentrations in different aliments.

This general approach, commonly used in nutritional epidemiology, allows for standardized comparisons across dietary groups, especially in the absence of individual-level exposure data. The mean value method is supported by previous research and has been adopted in similar contexts for modeling nutrient intake [8,9].

### 2.5. Data Preparation for Shannon Index Measurement

The calculation of the Shannon index value for each diet was performed through the execution of a mini-review, employing a designated search protocol and strategy across the two databases, Google Scholar and PubMed. The inclusion criteria were met by studies containing data on the specified diet and measurements of the Shannon index for various dietary patterns. A comprehensive collection of relevant studies was conducted, followed by a meticulous review and data extraction for the Shannon index across diverse dietary regimens.

### 2.6. The Relative Risk (RR) of Crohn’s Disease

To calculate the relative risk (RR) of Crohn’s disease in different countries, a comprehensive list of the prevalence of Crohn’s disease was prepared based on an epidemiological investigation. The relative risk (RR) of Crohn’s disease for each country was calculated as the ratio between the prevalence in the given country and the lowest observed prevalence among all analyzed nations [10]. The formula used wasRRi = Pi/Pmin
where

RRi is the relative risk of Crohn’s disease for country iii;Pi is the prevalence of the disease in country iii;Pmin is the lowest observed prevalence among all analyzed countries.

To obtain a standardized scale ranging from −1 to +1, the relative risk was normalized using the following formula:RRnorm,i = 2 × ((RRi − RRmin)/(RRmax − RRmin)) − 1
where

RRnorm,i is the normalized relative risk for country iii;RRmin and RRmax are the minimum and maximum observed RR values, respectively.

This transformation ensures that the lowest relative risk value is scaled to −1 and the highest value is scaled to +1.

## 3. Results

### 3.1. Estimation of S100B in Different Dietary Patterns

S100B concentration in diets from different geographical areas was estimated and compared with microbiota biodiversity data as reported from studies on healthy subjects. An analysis of S100B levels suggested a potential link between dietary patterns and eubiosis (Figure 2), indicating that diets rich in higher S100B concentrations, such as the Italian diet and Mediterranean diet, may mitigate dysbiosis processes. In Table 1, the estimated S100B presence in different diets (µg/kg) and Shannon alpha diversity (Shannon index) across different dietary patterns were reported.

The observed data highlight significant variations among diets, with the Italian diet showing the highest S100B concentrations (~95.08 µg/kg) and the Western diet (American-style diet) reporting the lowest (~49.54 µg/kg). Notably, the Okinawan diet and Korean diet show both low S100B concentrations (~51.53 µg/kg and ~49.23 µg/kg, respectively) and reduced microbiota diversity (Shannon index: 2.7 and 1.4). In contrast, the Mediterranean diet exhibits moderate S100B levels (~80.03 µg/kg) and relatively high Shannon indexes (~3.0), suggesting a consistent link between diet composition and gut microbiota diversity (Figure 2).

### 3.2. Epidemiological Investigation into Dietary Patterns with Different Levels of S100B

A total of 31 studies met the inclusion criteria, comprising over 1,000,000 participants (Appendix A, Appendix A). The included studies originated from diverse regions, with the majority conducted in Europe (61%), followed by North America (23%), Asia (13%), and other regions including Australia and Africa (3%) [11,12,13,14,15,16,17,18,19,20,21,22,23,24,25,26,27,28,29,30,31,32,33,34,35,36,37,38,39,40,41,42,43] (Appendix A). The Mediterranean diet was predominantly studied in Southern European countries such as Italy, Spain, and Greece, while the studies on plant-based diets were evenly distributed across continents, whereas Nordic diet studies were primarily conducted in Scandinavia (Appendix A). In analyses of data, the Mediterranean diet was associated with a 23% reduction in CVD risk (RR: 0.77, 95% CI: 0.71–0.83) and a 20% reduction in overall mortality (HR: 0.80, 95% CI: 0.74–0.87). Thai diets showed a 16% reduction in T2DM risk (RR: 0.84, 95% CI: 0.78–0.89) and were associated with lower BMI and improved glycemic control. The Nordic diet demonstrated a moderate reduction in inflammation markers and improved lipid profiles, though its effect on long-term chronic disease risk remains inconclusive (Data available in Appendix A). The pooled analysis of dietary patterns revealed that adherence to a Mediterranean diet significantly reduced the overall risk of chronic diseases (Figure 3A). The meta-regression indicated that dietary type was a significant predictor of disease risk reduction (*p* = 0.014), while geographic origin had a weaker influence (*p* = 0.067). Heterogeneity was moderate (I^2^ = 48%), suggesting some variation across study populations. The forest plot confirmed a consistent trend favoring whole-food, plant-rich diets in reducing chronic disease risk. Funnel plot analysis and Egger’s test (*p* = 0.032) indicated a low risk of bias publication (Appendix A). To investigate the potential relationship between S100B concentration (µg/kg) and Pooled Relative Risk (RR) across different dietary patterns, a meta-regression analysis was conducted using an ordinary least squares regression model (Appendix A–S4). The independent variable was the S100B concentration measured in various diets, while the dependent variable was the pooled RR extracted from the meta-analysis. The results indicate a negative association between S100B levels and RR (β = −0.0086), suggesting that lower S100B concentrations may be associated with higher relative risks for Chronic Disease. The regression coefficient for S100B is −0.002 (*p* = 0.302), suggesting that an increase in S100B levels is associated with a slight reduction in the relative risk of Chronic Disease (Figure 3A). However, the *p*-value is slightly above the conventional threshold (0.05), meaning that this effect is suggestive but not statistically significant at the 95% confidence level.

The forest plot illustrates the relative risk (RR) of Crohn’s disease across different countries (Figure 3B), based on the disease prevalence in each nation compared to the country with the lowest observed prevalence [42]. The results show a significant variation in relative risk across different regions worldwide. The Middle East exhibits the highest relative risk, indicating a significantly higher prevalence of Crohn’s disease in this region. The United States also shows high relative risk values, suggesting that the disease is more common in these populations compared to countries with lower prevalence rates [44,45]. Italy, France, the Nordic countries, and Thailand fall into an intermediate risk category, with prevalence levels that are notable but lower than in the USA and the Middle East [46,47,48,49]. Okinawa has a lower relative risk than Italy and France but remains higher than in countries with the lowest prevalence rates [50]. China, South Korea, India, Pakistan, Bangladesh, and West Africa have the lowest relative risk values, indicating a low prevalence of Crohn’s disease in these regions [44,51,52,53]. West Africa has the lowest relative risk, serving as the reference country for RR calculations. There is a higher prevalence of Crohn’s disease in Western industrialized countries (USA, France, Italy, and Nordic countries) compared to Asian and African nations. Diet, lifestyle, and genetic factors may contribute to this distribution, with an increased risk in industrialized nations compared to developing countries. When we analyzed these data with respect to the estimated presence of S100B in the diet, the meta-regression revealed a significant negative association between S100B concentration and normalized relative risk of Crohn’s disease (β = –0.0356, *p* < 0.001), suggesting that higher S100B levels are linked to lower disease risk (Figure 3B). The coefficient of determination (R^2^ = 0.780) suggests that approximately 78% of the variation in RR can be explained by S100B levels, though the small sample size limits the robustness of this finding. The regression coefficient for S100B is −0.002 (95% CI: −0.006 to 0.0019, *p* = 0.302), indicating a non-significant trend toward decreased chronic disease risk with increasing S100B. For Crohn’s disease, the coefficient of determination (R^2^ = 0.780, *p* < 0.001) suggests a robust inverse association between dietary S100B levels and disease prevalence across countries.

## 4. Discussion

The present epidemiological data support, on the concrete ground of human diets, previous results obtained in silico and on experimental animals, indicating that S100B, present in milk, dairy products, certain vegetables, and some fruits, may interact with microbiota biodiversity, resulting in a healthy feeding style [1]. The presence of S100B protein in different edible plants and dairy products opens new perspectives for its role in nutraceuticals as well as in dietary supplements, with the aim of supporting eubiosis, modulating gut microbiota, and potentially contributing to the prevention of chronic diseases [54,55,56,57,58]. The results emphasize the notable presence of S100B in different dietary patterns, its relationship with microbiota diversity, and its potential influence on health outcomes. The epidemiological study indicated that dietary patterns, especially Mediterranean and some vegetarian-based diets, are linked to a reduced risk of chronic diseases like cardiovascular disease (CVD). The differing presence of S100B in these diets indicates a putative connection between dietary consumption, modulation of the gut microbiota, and a decrease in systemic inflammation. Interestingly, the Italian diet, which reflected the estimated highest concentration of S100B (~95.08 µg/kg), also showed a comparatively elevated Shannon index value, suggesting a higher degree of microbial biodiversity, as already reported in several previous studies. This is consistent with earlier research highlighting the advantageous effects of eubiotic diets on both metabolic and neurological health. The Shannon index is a commonly used measure of alpha-diversity that accounts for both species richness and evenness in microbial communities. Higher Shannon index values indicate a more diverse and balanced microbiota, which is generally associated with ecological resilience and host health.

The meta-analysis investigating dietary patterns and chronic disease prevention supports the hypothesis that S100B may exert beneficial effects through modulation of the gut microbiota. A moderate positive association (R^2^ = 0.537, *p* < 0.05) was found between dietary S100B and microbial alpha-diversity (Shannon index). This supports the concept that S100B-rich diets may contribute to eubiosis and microbial ecosystem resilience, potentially via trophic and signaling interactions mediated by enteric glial cells [6]. S100B is a calcium-binding protein primarily secreted by astrocytes in the CNS and by enteric glial cells in the gastrointestinal tract. In peripheral tissues, including the gut, it participates in cell signaling, immune modulation, and epithelial barrier maintenance. Following dietary intake or local expression, S100B may influence gut microbial composition indirectly by acting on epithelial or glial targets, or by generating bioactive fragments through proteolytic cleavage.

The forest plot analysis (Figure 3B) confirms that Crohn’s disease is significantly more prevalent in Western and industrialized countries, whereas it has a much lower prevalence in Asian and African countries. These findings further suggest that environmental, dietary, and genetic factors may influence disease distribution, even if this is within a complex network of risk factors and confounding factors. The most striking result in S100B association is when comparing some well-established or extreme values in the epidemiological scenarios, such as the high prevalence in the Middle East and the USA with respect to other regions worldwide. Overall, the research offers new insights into the dietary origins of S100B, its connection with diversity in the gut microbiota, and its potential role in preventing chronic diseases. These results bolster the idea that the dietary intake of S100B might significantly impact gut–microbiota interactions and overall health, highlighting the need for further research into its function as a novel component or marker in different dietary styles. While the precise mechanisms that may influence microbiota biodiversity are still under investigation, several plausible hypotheses can be proposed. S100B is a calcium-binding protein with known trophic and immunomodulatory effects in neural tissues at nanomolar concentrations [4], and similar actions might occur in the gastrointestinal tract, particularly via enteric glial cells, which are known to interact with the gut microbiota [59]. S100B could potentially act as a signaling molecule, influencing epithelial barrier function or local immune responses, both of which are critical for microbial composition and stability [55]. Additionally, it is hypothesized that S100B might undergo proteolytic degradation in the gut lumen, producing bioactive peptides or amino acid fragments that can serve as metabolic substrates for specific bacterial taxa. Some in silico predictions have identified cleavage motifs in the S100B sequence that may be targeted by bacterial proteases [5], raising the possibility of a direct interaction between S100B and microbiota metabolism. These scenarios suggest that the presence of S100B in the diet could contribute to the selective enrichment of beneficial microbial populations or promote overall microbial diversity. Further studies, including in vitro fermentation models and metagenomic profiling, are needed to test these hypotheses and clarify the mechanistic role of dietary S100B in shaping gut microbiota. Crohn’s disease was selected as a relevant inflammatory condition because it is consistently associated with gut dysbiosis, and particularly reduced microbial alpha-diversity. Moreover, S100B is known to be overexpressed in inflamed intestinal tissues and may contribute to immune dysregulation through the RAGE pathway. For these reasons, Crohn’s disease represents a biologically plausible model to explore how dietary S100B intake could influence microbiota-driven inflammation. Increasing microbial diversity is considered beneficial in this context, as it supports intestinal barrier integrity, modulates immune responses, and enhances the production of anti-inflammatory metabolites such as short-chain fatty acids—all of which are protective factors in Crohn’s disease.

Beyond its potential protective role in inflammatory bowel diseases, S100B has been implicated in other chronic disorders, including obesity, type 2 diabetes mellitus (T2DM), and various neurological and neuroinflammatory conditions. Elevated endogenous levels of S100B have been observed in patients with obesity and insulin resistance, where they may contribute to systemic low-grade inflammation and metabolic dysfunction [60]. Similarly, alterations in S100B expression have been reported in Alzheimer’s disease and depression, where gut microbiota dysbiosis has also been recognized as a contributing factor [61,62]. These findings suggest a broader role for S100B–microbiota interactions across multiple chronic disease pathways. Although the present study focused on Crohn’s disease, the observed association between higher dietary S100B levels and increased microbial diversity may have broader implications for both metabolic and neurological health, particularly in conditions where eubiosis is known to exert a protective effect. Notably, S100B is also known to interact with the receptor for advanced glycation end-products (RAGE), a pattern recognition receptor involved in inflammatory signaling and intestinal barrier regulation. Activation of the S100B–RAGE axis has been associated with either pro- or anti-inflammatory responses depending on the context and concentration, and may represent a plausible mechanism linking dietary S100B intake with modulation of gut inflammation, including in Crohn’s disease [63,64]. While the precise mechanisms that may influence microbiota biodiversity are still under investigation, several plausible hypotheses can be proposed to explain how S100B, despite its relatively low dietary concentration, may impact gut microbial ecology. Protein-derived compounds can influence microbiota through nitrogen supply, the generation of bioactive peptides, or molecular signaling. S100B, in particular, may exert effects via enteric glial interactions, modulation of the epithelial barrier, or even microbial enzymatic cleavage, producing trophic fragments that influence community dynamics [65]. Secondly, the quantification of S100B in food items was derived from compositional datasets that may be limited by variability in food origin, processing methods, and analytical sensitivity. This variability in S100B concentrations—driven by animal species, lactation stage, diet, breed, health status, and analytical techniques (e.g., ELISA vs. Western blot)—has been documented previously [1,66]. We acknowledge this heterogeneity and have addressed it by using mean values in our estimates. While this represents a simplification, it is a justified and commonly accepted method in large-scale dietary studies, particularly when dealing with food composition data and population-level patterns. These biological and methodological differences are consistent with previously reported heterogeneity in milk composition due to species (cow, goat, sheep), diet, health status, and lactation stage [67], as well as variability introduced by different analytical techniques such as ELISA versus Western blot [68]. Such factors are also consistent with the findings discussed by [1] Michetti et al. (2025) regarding the trophic dietary role of S100B. The dietary intake estimates did not account for bioavailability or the possible degradation of S100B during food preparation or digestion, which could affect its functional impact on the microbiota [56]. Moreover, the gut microbiota diversity was assessed using population-level Shannon indices, which may not fully capture individual variability or the functional capabilities of specific microbial taxa. Longitudinal studies with individual-level microbiome sequencing and controlled dietary interventions would be needed to clarify the mechanistic role of S100B. Finally, although the hypothesis that dietary S100B can influence the gut–brain axis and inflammation is biologically plausible, it remains speculative without direct mechanistic or interventional evidence in humans. An interesting and consistent trend emerges when examining S100B concentrations across different diets, whether derived from plant-based foods or dairy products. A further limitation is the absence of in vivo validation, which would be necessary to confirm the potential biological effects of dietary S100B on microbiota composition and function. Future studies should investigate this relationship using controlled animal models or clinical dietary interventions, to establish causality and explore underlying mechanisms. A key limitation is the lack of adjustment for potential confounders such as dietary fiber intake, antibiotic exposure, or physical activity, which are known to impact microbiota diversity. Future studies should include multivariate regression models with individual-level data. The novelty of this study lies in its integrative approach, combining compositional food analysis, microbiome ecology, and epidemiological modeling to explore a previously unaddressed hypothesis: the potential role of dietary S100B in modulating gut microbiota and influencing chronic disease risk. While the findings are promising, we emphasize that this study is exploratory and intended to generate hypotheses. The observed associations between dietary S100B and microbiota diversity cannot establish causality. Rather, our results provide a conceptual framework for future investigations. Controlled clinical trials, longitudinal studies, and mechanistic experiments—including gnotobiotic animal models and targeted dietary interventions—will be essential to elucidate the potential role of dietary S100B in shaping the gut microbiome and promoting health.

## 5. Mechanistic Considerations and Future Research Directions

This paper presents a novel perspective for the putative influence of S100B dietary uptake on health, opening possible strategies for chronic disease prevention. The general approach is based on data available from the scientific literature, experimental investigations, and epidemiological analysis. The hypothesis concerns a mechanism involving eubiosis and the gut microbiota axis. While requiring further investigation, a lot of the evidence came from combining the results reported in different experimental studies by different methodologies: metanalysis, in silico by bioinformatics, in vivo studies on mice, and in vitro using immunodetection of S100B in different food sources. The observation of a nutritional role for S100B is enforced by its detection in milk, as a first trophic component in the human breastfeeding diet. Here, we suppose the possibility of continuing S100B uptake in an adult diet as well, mainly through plants and dairy products, contributing to establishing a healthy microbiota. Even if S100B was shown to affect gut microbiota biodiversity in mice and several lines of evidence support this effect, however, the detailed mechanisms need to be unraveled. Potential pathways through which dietary S100B might influence the gut microbiota biodiversity in human populations may include (i) the capability of this HLH-protein to interact with bacterial and/or human ligands through a knock–fist domain; (ii) the modulation of enteric glial cells, which are vital for maintaining gut barrier integrity and immune regulation; (iii) direct interactions with gut epithelial or immune cells via RAGE or other receptors, which could affect cytokine release and inflammatory pathways; (iv) indirect action as a trophic factor or signaling molecule, influencing microbial community dynamics. Nevertheless, these and other mechanistic pathways remain speculative, and the epidemiological approach based on a metanalysis and systematic review of available studies is limited by potential confounding factors like dietary fiber intake, overall diet quality, and lifestyle choices, demanding additional investigations involving large cohort studies. While the findings are promising, we emphasize that this study is exploratory and intended to generate hypotheses. The observed associations between dietary S100B and microbiota diversity cannot establish causality. Rather, our results provide a conceptual framework for future investigations. Controlled clinical trials, longitudinal studies, and mechanistic experiments—including gnotobiotic animal models and targeted dietary interventions—will be essential to elucidate the potential role of dietary S100B in shaping the gut microbiome and promoting health.

## 6. Conclusions

This study offers preliminary evidence that dietary intake of S100B—a protein found in dairy products, certain vegetables, and fruits—may be associated with increased gut microbiota diversity, a feature commonly linked to health resilience and reduced chronic disease risk. The analysis suggests that dietary patterns rich in S100B, such as Mediterranean and Italian diets, are correlated with higher alpha-diversity and lower estimated risk of inflammatory conditions such as Crohn’s disease. These results support the biological plausibility of S100B as a modulator of the gut ecosystem, potentially acting through immunomodulatory and trophic mechanisms. However, as this study is exploratory and based on indirect estimations, further research is essential. Future studies should focus on confirming these associations through mechanistic experiments, controlled dietary interventions, and longitudinal population studies. This work provides a conceptual basis for integrating protein-derived bioactive molecule like S100B into the broader field of chronic disease prevention.

## Figures and Tables

**Figure 1 biomolecules-15-01047-f001:**
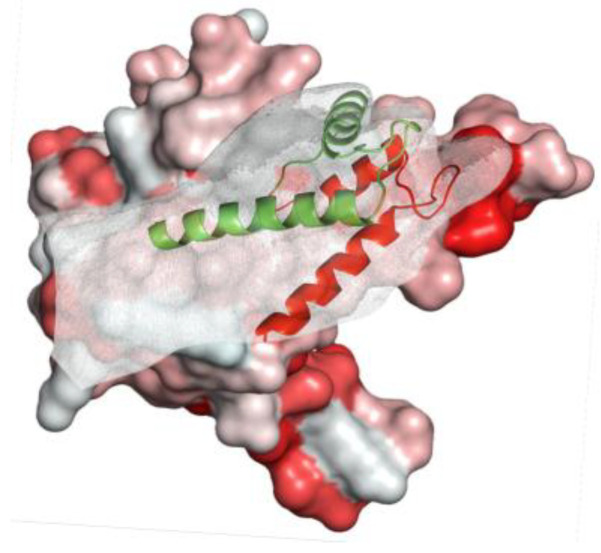
S100B consists of 81 amino acids organized into two functional domains: the S100 domain (amino acids 4–46) and the EF-hand domain (residues 53–81). Its three-dimensional structure has been described as resembling a ‘knock-fist’ shape, which may facilitate protein–protein interactions. In this model, the S100B domain forms a predominantly hydrophobic region (‘fist’), while the protruding area (‘knock’) is characterized by polar, hydrophilic residues [3,5].

**Figure 2 biomolecules-15-01047-f002:**
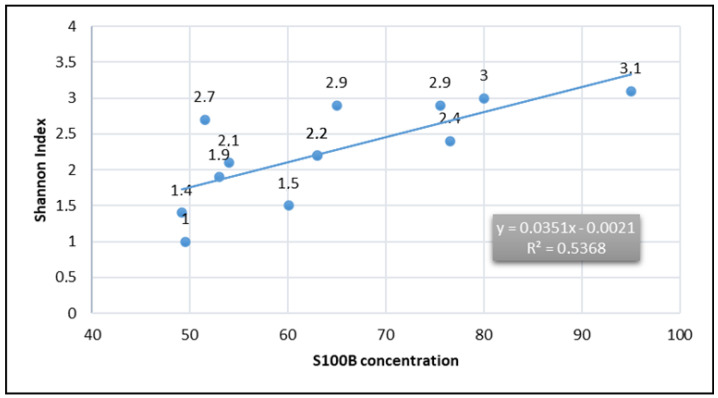
Correlation between S100B and Shannon index values was observed across different diets worldwide. A moderate positive correlation was found (R² = 0.537, *p* < 0.05), suggesting that higher dietary concentrations of S100B are associated with increased alpha-biodiversity, supporting a potential eubiotic role.

**Figure 3 biomolecules-15-01047-f003:**
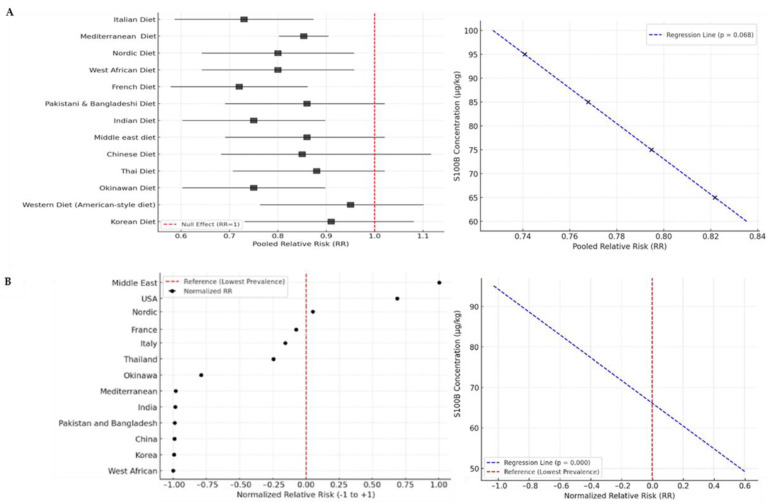
Epidemiological analysis of S100B in dietary patterns. (**A**) Forest plot displaying the pooled relative risk (RR) estimates for chronic diseases in different dietary patterns. Each square represents the weighted RR for a specific dietary pattern, with the size of the square indicating the study’s weight in the meta-analysis. The horizontal lines represent 95% confidence intervals (CI), while the vertical red dashed line at RR = 1 indicates no effect. The findings highlight the significant protective effects of the Mediterranean and plant-based diets in reducing chronic disease risk. The meta-regression curve illustrates the slight trend of association between S100B levels and the relative risk of chronic diseases (*p* = 0.068). The curve shows a negative trend, suggesting that higher S100B levels may be associated with lower disease risk across dietary patterns. Although the relationship is not statistically significant, it indicates a potential mechanistic link worth further investigation. (**B**) Forest plot displaying the normalized relative risk (RR) estimates for Crohn’s disease across different countries. Each circle represents the RR. A red dashed vertical line marks the reference value, corresponding to the lowest observed prevalence [10]. Countries with higher RR values, such as the Middle East and the USA, show a significantly increased prevalence of Crohn’s disease compared to regions with lower RR values, such as China, India, and West Africa. The distribution suggests a higher burden of the disease in Western and industrialized countries. The curve shows a significant negative trend, suggesting that higher S100B levels may be associated with lower Crohn’s disease risk (*p* < 0.001). This statistically significant relationship indicates a potential mechanistic link that warrants further investigation.

**Table 1 biomolecules-15-01047-t001:** Comparative analysis of S100B concentration with Shannon index.

Type of Diet	S100B Concentration (µg/kg)	Shannon Index
Italian	95.08 ± 0.23	3.1
Mediterranean	80.03 ± 0.39	3.0
Nordic	76.53 ± 0.32	2.4
West African	75.54 ± 0.32	2.9
French	65.03 ± 0.18	2.9
Pakistani and Bangladeshi	63.03 ± 0.37	2.2
Indian	63.03 ± 0.37	2.2
Middle east	60.08 ± 0.22	1.5
Chinese	54.03 ± 0.16	2.1
Thai	53.03 ± 0.16	1.9
Okinawan	51.53 ± 0.46	2.7
Western (American style)	49.54 ± 0.43	1.0
Korean	49.23 ± 0.02	1.4

## Data Availability

The data supporting the findings of this study are available from the corresponding author upon reasonable request.

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
