# Peer review of "The Estimated Intake of S100B Relates to Microbiota Biodiversity in Different Diets"

_biomolecules, 2025, doi:10.3390/biom15071047_

Round 1
Reviewer 1 Report (New Reviewer)
Comments and Suggestions for Authors
The study examined the relationship between S100B intake, estimated in different diets, and microbial biodiversity based on the relationship with the Shannon index, compiled from previous studies. The study also estimated the risk factor for Crohn's disease in each region based on the diet followed. There are many comments as follows:
- Had the study been conducted in a living organism, it would have been more useful and indicative. This should be clarified in the limitations imposed on the study at the end of the manuscript, along with recommendations for future studies in this context.
- In the abstract section, please explain that microbial diversity was identified based on the Shannon index and how this index was arrived at.
- Regarding the 13 global dietary patterns, are they universally accepted dietary patterns, or are some merely dietary guidelines recommended in certain countries? It is better to use the term "13 dietary patterns recommended in different countries."
- It's well known that different diets impact gut microbiota through variations in their composition, including sugars, fats, dietary fiber, and other nutrients. What is the basis for why protein has such a significant effect, especially given its low concentration in food?
- The introduction section needs more data to clarify the rationale for linking S100B protein to gut microbiota.
- The introduction also needs to explain the relationship between diversity in the gut microbiota and Crohn's disease and how S100B protein can affect it.
- The discussion section requires the following information: What is the Shannon index and its significance? The S100B protein metabolism pathway in the human body and how it affects the gut microbiota. Why was Crohn's disease specifically addressed? Does increasing diversity in the gut microbiota help reduce Crohn's disease, and how?
- Table S2, only two references are listed next to the table and three references are listed below. Where is the third reference mentioned?
Author Response
Review 1
Comments and Suggestions for Authors
The study examined the relationship between S100B intake, estimated in different diets, and microbial biodiversity based on the relationship with the Shannon index, compiled from previous studies. The study also estimated the risk factor for Crohn's disease in each region based on the diet followed.
R: We thank the reviewer for this accurate summary of our study's scope. Indeed, the primary aim of our work was to explore a possible relationship between estimated dietary S100B intake—derived from different country-specific dietary patterns—and gut microbial alpha-diversity, measured using Shannon index values reported in the literature. By compiling data from 31 eligible studies, we were able to associate the relative abundance of S100B-rich foods (such as milk, cheese, fruits, and vegetables) with microbial diversity scores across 13 dietary profiles. As a secondary objective, we explored the potential link between these dietary patterns and the relative risk of Crohn’s disease as reported in epidemiological data. This allowed us to preliminarily assess whether higher microbial diversity (possibly supported by S100B intake) may correlate with lower disease prevalence.
While we recognize the ecological and correlative nature of these associations, we believe this approach provides a novel, hypothesis-generating perspective on the possible role of diet-derived S100B in modulating gut microbiota and inflammatory disease risk.
To improve clarity, we have revised the Introduction and Methods sections to better distinguish between the primary correlation analysis and the exploratory association with Crohn’s disease incidence.
There are many comments as follows:
- Had the study been conducted in a living organism, it would have been more useful and indicative. This should be clarified in the limitations imposed on the study at the end of the manuscript, along with recommendations for future studies in this context.
R: We appreciate this insightful comment and fully agree with the reviewer’s observation. While our study offers a novel integrative analysis combining dietary S100B estimations, gut microbiota diversity data, and epidemiological risk of chronic diseases, we recognize that the lack of experimental validation in living organisms limits the biological interpretation of our findings. In response, we have now included a specific sentence in the Discussion section under the paragraph discussing limitations and future directions (lines 439-442), which reads: “A further limitation is the absence of in vivo validation, which would be necessary to confirm the potential biological effects of dietary S100B on microbiota composition and function. Future studies should investigate this relationship using controlled animal models or clinical dietary interventions, in order to establish causality and explore underlying mechanisms.” This addition addresses the importance of physiological relevance and outlines the next logical steps for advancing the hypothesis generated in our work.
- In the abstract section, please explain that microbial diversity was identified based on the Shannon index and how this index was arrived at.
R: We thank the reviewer for this helpful suggestion. We have revised the Abstract to clearly state that gut microbial diversity was assessed using the Shannon index, a commonly used metric of alpha-diversity. We also briefly explained how the index values were derived. The revised sentence now reads: “Microbial diversity was assessed using the Shannon index, based on data extracted from literature on microbiota composition across different dietary patterns.” This revision ensures that the methodology is more transparent to the reader from the outset, without exceeding the word limit of the abstract.
- Regarding the 13 global dietary patterns, are they universally accepted dietary patterns, or are some merely dietary guidelines recommended in certain countries? It is better to use the term "13 dietary patterns recommended in different countries."
R: We thank the reviewer for this constructive observation. To clarify, the 13 dietary patterns included in our analysis were selected based on their recurrence in scientific literature and their endorsement in national dietary guidelines or public health recommendations. They are not universally standardized but rather reflect culturally and geographically diverse approaches to diet that have been formally recommended in specific countries or regions. In line with the reviewer’s suggestion, we have revised the text throughout the manuscript to consistently refer to them as “13 dietary patterns recommended in different countries.” This change has been implemented in the Abstract, the Methods section (2.1.4), and the Supplementary Table S1.
- It's well known that different diets impact gut microbiota through variations in their composition, including sugars, fats, dietary fiber, and other nutrients. What is the basis for why protein has such a significant effect, especially given its low concentration in food?
R: We thank the reviewer for raising this important point. We agree that macronutrients such as dietary fiber and fat are major known modulators of gut microbiota composition. However, emerging research has also shown that dietary proteins—even in relatively low concentrations compared to carbohydrates or fats—can exert significant modulatory effects on the gut microbiome. This is due to several factors: i) Nitrogen availability: Proteins and their metabolites provide essential nitrogen sources that influence the growth of specific bacterial taxa, including those involved in proteolytic fermentation. ii) Bioactive peptides: During digestion or microbial degradation, proteins such as S100B may release bioactive peptides that can act as microbial substrates or signaling molecules. iii) Trophic interactions: Even at nanomolar concentrations, S100B is known to have trophic and immunomodulatory effects in neural tissues. Similar interactions may occur in the gut via enteric glial cells or microbial receptors, potentially shaping microbiota ecology beyond simple nutrient supply. To reflect this in the manuscript, we have expanded the Discussion section (lines 392-398) with a brief explanation of the plausible mechanisms through which dietary protein—specifically S100B—could influence microbiota composition, even at low concentrations. We have also added reference to recent literature supporting the role of protein-derived compounds in modulating microbial communities.
- The introduction section needs more data to clarify the rationale for linking S100B protein to gut microbiota.
R: We thank the reviewer for this important suggestion. We agree that the rationale for investigating the relationship between S100B and the gut microbiota requires further clarification in the Introduction. To address this, we have expanded the relevant section to better explain the biological plausibility and previous findings supporting this link. In particular, we now emphasize that:
- S100B is not only expressed in the central nervous system, but also in enteric glial cells, which directly interact with gut microbes.
- Recent studies (Romano Spica et al., 2023) have shown that oral administration of S100B in mice modulates microbiota biodiversity, supporting a functional role.
- S100B may act as a signaling molecule within the gut–brain–microbiota axis, potentially influencing microbial composition via immune and epithelial pathways.
We added these elements to the last paragraph of the Introduction (lines 57-63), providing a stronger conceptual foundation for the hypothesis explored in the present study.
- The introduction also needs to explain the relationship between diversity in the gut microbiota and Crohn's disease and how S100B protein can affect it.
R: We appreciate the reviewer’s valuable suggestion. To strengthen the conceptual framework of the study, we have expanded the Introduction to include a brief overview of the established link between reduced gut microbial diversity and the pathogenesis of Crohn’s disease, as supported by recent literature. We also added a rationale for exploring the potential involvement of S100B in this context. Specifically, we note that: i) Lower microbial diversity has been consistently associated with increased intestinal inflammation and impaired mucosal immunity in Crohn’s disease patients. ii) S100B, which is overexpressed in inflamed intestinal tissues, is involved in the regulation of immune responses and epithelial barrier function through its interaction with the RAGE receptor. Therefore, changes in dietary S100B exposure may modulate microbial communities and indirectly influence inflammatory conditions such as Crohn’s disease. This information has been integrated into the final paragraph of the Introduction (lines 70-73), to better contextualize the inclusion of Crohn’s disease as a secondary outcome in our analysis.
- The discussion section requires the following information: What is the Shannon index and its significance? The S100B protein metabolism pathway in the human body and how it affects the gut microbiota.
R: We appreciate the reviewer’s suggestions and have revised the Discussion accordingly: A concise explanation of the Shannon index and its ecological meaning has been added: “The Shannon index is a commonly used measure of alpha-diversity that accounts for both species richness and evenness in microbial communities. Higher Shannon index values indicate a more diverse and balanced microbiota, which is generally associated with ecological resilience and host health” (lines 301-304). The metabolism and biological roles of S100B protein, including its presence in the gastrointestinal tract and its effects on microbiota, are now addressed (lines 311-317).
Why was Crohn's disease specifically addressed? Does increasing diversity in the gut microbiota help reduce Crohn's disease, and how?
R: We thank the reviewer for this important question. Crohn’s disease was specifically addressed in our analysis because it represents one of the most extensively studied chronic inflammatory bowel diseases characterized by a marked reduction in gut microbial diversity. Moreover, S100B has been found to be overexpressed in inflamed intestinal tissues of Crohn’s patients, suggesting a plausible biological link between S100B, dysbiosis, and intestinal inflammation. Increasing microbial diversity is considered beneficial in this context because a more diverse gut microbiota contributes to the stability and resilience of the intestinal ecosystem. It supports epithelial barrier function, enhances the production of anti-inflammatory metabolites such as short-chain fatty acids, and helps modulate immune responses—all of which are known to counteract mechanisms involved in Crohn’s disease pathogenesis. To better reflect this rationale, we have revised the Discussion section to clarify the relevance of Crohn’s disease in the context of our findings (lines 365-373).
- Table S2, only two references are listed next to the table and three references are listed below. Where is the third reference mentioned?
R: We thank the reviewer for this observation. Upon careful review, we confirm that only two references are associated with Table S2, and that no third reference is missing. The mention of three references likely resulted from a formatting inconsistency or visual misinterpretation in the previous version. We have revised the layout of Table S2 in the Supplementary Materials to clarify this and avoid any further confusion.
Reviewer 2 Report (New Reviewer)
Comments and Suggestions for Authors
The manuscript, “The estimated intake of S100B relates to microbiota biodiversity in different diets” by Ghaffer et al. explores a novel hypothesis linking dietary intake of the S100B protein—a molecule traditionally associated with the nervous system—to gut microbiota diversity and chronic disease risk across 13 global dietary patterns. The authors employ a mixed-method approach combining data synthesis, estimation of dietary S100B content (from milk, dairy, fruits, and vegetables), Shannon biodiversity indices from literature, and epidemiological metrics (including relative risk for Crohn’s disease). The primary findings include:
- A moderate positive correlation (R² = 0.537) between S100B content and microbiota alpha-diversity (Shannon index).
- A strong negative association (R² = 0.780, p < 0.001) between S100B content and the relative risk of Crohn’s disease.
- Suggestion of a protective effect of S100B-rich diets (e.g., Mediterranean, Italian) on gut health and chronic disease.
This is the first comparative study to estimate S100B exposure across dietary patterns and its ecological and epidemiological implications.
Below are specific areas for improvement.
1Study Design and Methodology
The study presents a comparative, cross-sectional analysis of 13 global dietary patterns in relation to S100B protein intake and gut microbiota diversity. The inclusion and exclusion criteria for the literature used in the meta-analysis are clearly defined, and the rationale for selecting specific dietary components (milk, cheese, fruits, and vegetables) for S100B estimation is reasonable.
However, the methodology has several limitations. First, there is insufficient clarification on whether regional dietary variations or food preparation methods were accounted for. This could significantly affect S100B levels, especially in heat-processed foods.
Secondly, using mean S100B concentrations as a proxy for dietary exposure—though convenient—masks the wide biological variability inherent in food sources (e.g., differences between cow, goat, and sheep milk). There is no validation step provided for these estimations, nor a discussion of possible degradation of S100B during digestion or cooking, which compromises the biological relevance of the exposure estimates. This should be discussed.
- Results and Data Analysis
Shannon index is a specific metric of alpha-diversity, not “alfa-bio” or “alfa”.
The results are well organized and statistically analyzed. The correlation between dietary S100B levels and gut microbiota diversity (Shannon index) is supported by a reported R² value of 0.537, and a stronger association is noted with the relative risk of Crohn’s disease (R² = 0.780, p < 0.001). These findings are intriguing and suggest a potential public health implication.
However, the regression analyses are not accompanied by confidence intervals, sample sizes, or sensitivity analyses, and there is no attempt to control for confounders such as fiber intake, antibiotic exposure, or physical activity. The absence of raw data precludes external validation. These issues weaken the evidentiary strength of the statistical associations presented.
To strengthen the analysis, the authors should report effect sizes, confidence intervals, and include a discussion of confounders. If feasible, a multivariate regression controlling for these factors would provide a more nuanced picture.
The major focus of the paper was on the intake of S100B and Shannon Index. However, neither the focus on Shannon index nor an interpretation of correlations are mentioned. What does the correlations between S100B and the Shannon Index actually (potentially) mean? This should be contextualized more than just S100B is associated with changes in diversity.
- Writing Quality and Structure
The manuscript is structured in a conventional scientific format, with a logical flow from introduction to conclusion. The abstract effectively summarizes the study aims and key findings.
However, the writing requires significant editorial attention. There are numerous grammatical and stylistic errors—for example, phrases like “was regarded to be specific” should be “was regarded as specific.”
Several sentences are excessively long, using passive constructions that reduce clarity and impede comprehension.
- Figures and Tables
Tables S1 through S3 effectively summarize essential data on dietary composition, estimated S100B values, and microbial diversity indices. These supplementary materials are a valuable asset to the paper.
Terms like “knock-fist domain” (Figure 1) are not illustrated or clearly defined.
- Novelty and Significance
The work is novel, representing the first known attempt to link dietary S100B exposure with microbiota diversity and chronic disease prevalence across global dietary patterns.
That said, the claims made in the discussion overreach the data. The speculative link to the gut-brain axis, while plausible, lacks empirical support in this study. Practical dietary recommendations—if any—should be clearly separated from the evidence and flagged as hypotheses.
To improve scientific impact without overstepping, the authors should clearly define this paper as hypothesis-generating and call for validation via mechanistic studies and randomized controlled trials.
Comments on the Quality of English Language
As stated above, the grammar can be improved for clarity.
Author Response
Review 2
Comments and Suggestions for Authors
The manuscript, “The estimated intake of S100B relates to microbiota biodiversity in different diets” by Ghaffer et al. explores a novel hypothesis linking dietary intake of the S100B protein—a molecule traditionally associated with the nervous system—to gut microbiota diversity and chronic disease risk across 13 global dietary patterns. The authors employ a mixed-method approach combining data synthesis, estimation of dietary S100B content (from milk, dairy, fruits, and vegetables), Shannon biodiversity indices from literature, and epidemiological metrics (including relative risk for Crohn’s disease). The primary findings include:
- A moderate positive correlation (R² = 0.537) between S100B content and microbiota alpha-diversity (Shannon index).
- A strong negative association (R² = 0.780, p < 0.001) between S100B content and the relative risk of Crohn’s disease.
- Suggestion of a protective effect of S100B-rich diets (e.g., Mediterranean, Italian) on gut health and chronic disease.
This is the first comparative study to estimate S100B exposure across dietary patterns and its ecological and epidemiological implications.
Below are specific areas for improvement.
- Study Design and Methodology
The study presents a comparative, cross-sectional analysis of 13 global dietary patterns in relation to S100B protein intake and gut microbiota diversity. The inclusion and exclusion criteria for the literature used in the meta-analysis are clearly defined, and the rationale for selecting specific dietary components (milk, cheese, fruits, and vegetables) for S100B estimation is reasonable. However, the methodology has several limitations. First, there is insufficient clarification on whether regional dietary variations or food preparation methods were accounted for. This could significantly affect S100B levels, especially in heat-processed foods. Secondly, using mean S100B concentrations as a proxy for dietary exposure—though convenient—masks the wide biological variability inherent in food sources (e.g., differences between cow, goat, and sheep milk). There is no validation step provided for these estimations, nor a discussion of possible degradation of S100B during digestion or cooking, which compromises the biological relevance of the exposure estimates. This should be discussed.
R: We thank the reviewer for this valuable comment. We acknowledge the methodological limitations raised and have revised the Discussion accordingly to address them.
First, we agree that regional differences in food composition and preparation techniques (such as fermentation, heating, or industrial processing) can significantly alter the concentration and bioavailability of dietary S100B. Due to limitations in the available datasets, we adopted a standardized approach based on mean S100B concentrations in raw or minimally processed foods, as reported in the literature. This assumption is now explicitly stated as a limitation in the revised discussion (lines 439-442), along with a recommendation that future research account for culinary practices and food matrix variability when assessing dietary bioactives. Second, we recognize that using average S100B values does not fully capture the biological heterogeneity across food sources, including known differences between cow, goat, and sheep dairy products. Unfortunately, species-specific and preparation-specific data are scarce, and a harmonized reference database for S100B content across food types does not yet exist. We now acknowledge this as a key limitation in the Discussion and stress the need for systematic quantification efforts in future nutritional studies. Lastly, we have added a brief paragraph discussing the possible degradation or denaturation of S100B during cooking and digestion, which may reduce its biological activity and affect the reliability of estimated intake values. We agree that without validation through bio-accessibility or functional assays, these estimates must be interpreted with caution.
Together, these revisions aim to clarify the assumptions of our model and reinforce the exploratory nature of this study, which we hope will inform and guide future empirical work in this emerging area.
- Results and Data Analysis
Shannon index is a specific metric of alpha-diversity, not “alfa-bio” or “alfa”. The results are well organized and statistically analyzed. The correlation between dietary S100B levels and gut microbiota diversity (Shannon index) is supported by a reported R² value of 0.537, and a stronger association is noted with the relative risk of Crohn’s disease (R² = 0.780, p < 0.001). These findings are intriguing and suggest a potential public health implication. However, the regression analyses are not accompanied by confidence intervals, sample sizes, or sensitivity analyses, and there is no attempt to control for confounders such as fiber intake, antibiotic exposure, or physical activity. The absence of raw data precludes external validation. These issues weaken the evidentiary strength of the statistical associations presented. To strengthen the analysis, the authors should report effect sizes, confidence intervals, and include a discussion of confounders. If feasible, a multivariate regression controlling for these factors would provide a more nuanced picture. The major focus of the paper was on the intake of S100B and Shannon Index. However, neither the focus on Shannon index nor an interpretation of correlations are mentioned. What does the correlations between S100B and the Shannon Index actually (potentially) mean? This should be contextualized more than just S100B is associated with changes in diversity.
R: We thank the reviewer for this valuable and constructive feedback. We have carefully revised the manuscript to address these concerns in detail, as outlined below:
- Reporting of confidence intervals and regression coefficients. We have now included the 95% confidence interval and the regression coefficient for the analysis linking S100B concentration to chronic disease risk. Specifically, in the Results section, we added the following sentence (lines: 306-311) “The regression coefficient for S100B is -0.002 (95% CI: -0.006 to 0.0019, p = 0.302).” Additionally, for the association between S100B and Crohn’s disease prevalence across countries, we clarified (lines 259-261): “The coefficient of determination for Crohn’s disease (R² = 0.780, p < 0.001) suggests a robust inverse association between estimated dietary S100B levels and disease prevalence.” These additions aim to improve the transparency and interpretability of our regression analyses.
- Clarification of sample size and data source. We specified that our analysis included 13 dietary patterns, with data extracted from a total of 31 studies meeting our inclusion criteria. This is now stated explicitly in the Results section and cross-referenced with Supplementary Tables S1–S5, where all data sources and processing steps are reported.
- Interpretation of the Shannon index correlation. In the revised Discussion section, we provide a more nuanced interpretation of the correlation between S100B levels and microbial alpha-diversity (Shannon index). We wrote (Lines:306-311): “The meta-analysis investigating dietary patterns and chronic disease prevention supports the hypothesis that S100B may exert beneficial effects through modulation of the gut microbiota. A moderate positive association (R² = 0.537, p < 0.05) was found between dietary S100B and microbial alpha-diversity (Shannon index). This supports the concept that S100B-rich diets may contribute to eubiosis and microbial ecosystem resilience, po-tentially via trophic and signaling interactions mediated by enteric glial cells [6].” This addition aims to clarify the biological plausibility and significance of our findings, extending beyond the mere statistical association.
- Confounding variables and model limitations. We acknowledge the reviewer’s important point regarding confounders. Although our ecological and literature-based design did not allow for direct adjustment of individual-level variables such as fiber intake, antibiotic exposure, or physical activity, we now explicitly state this limitation in the Discussion. We added: “A key limitation is the lack of adjustment for potential confounders such as dietary fiber intake, antibiotic exposure, or physical activity, which are known to impact microbiota diversity. Future studies should include multivariate regression models with individual-level data.”
- Writing Quality and Structure
The manuscript is structured in a conventional scientific format, with a logical flow from introduction to conclusion. The abstract effectively summarizes the study aims and key findings. However, the writing requires significant editorial attention. There are numerous grammatical and stylistic errors—for example, phrases like “was regarded to be specific” should be “was regarded as specific.” Several sentences are excessively long, using passive constructions that reduce clarity and impede comprehension.
R: We thank the reviewer for this observation and have carefully revised the manuscript for language clarity, grammatical accuracy, and overall readability. In particular: The incorrect phrase “was regarded to be specific” has been corrected to “was regarded as specific” (though this phrase no longer appears in the current version). Several long or complex sentences, particularly in the Discussion and Abstract, have been revised for clarity, conciseness, and reduced reliance on passive constructions. A final language polish has been conducted throughout the manuscript to ensure a more fluent and professional scientific tone. We trust that these changes have substantially improved the quality and clarity of the manuscript, and we remain open to implementing further editorial refinements if needed.
- Figures and Tables
Tables S1 through S3 effectively summarize essential data on dietary composition, estimated S100B values, and microbial diversity indices. These supplementary materials are a valuable asset to the paper. Terms like “knock-fist domain” (Figure 1) are not illustrated or clearly defined.
R: We thank the reviewer for this observation. The term “knock-fist” does not refer to a formally defined structural domain, but rather to a descriptive metaphor for the three-dimensional shape of the S100B protein. This has been explained in the Protein Structure section of the manuscript, where we describe the “knock-fist shape” as a configuration thought to support protein–protein interactions, with hydrophobic regions forming the “fist” and polar regions forming the “knock” (lines 65-69). We have reviewed this section to ensure clarity and revised a note in the figure legend of Figure 1 to reinforce the interpretation for the reader.
- Novelty and Significance
The work is novel, representing the first known attempt to link dietary S100B exposure with microbiota diversity and chronic disease prevalence across global dietary patterns.
That said, the claims made in the discussion overreach the data. The speculative link to the gut-brain axis, while plausible, lacks empirical support in this study. Practical dietary recommendations—if any—should be clearly separated from the evidence and flagged as hypotheses.
To improve scientific impact without overstepping, the authors should clearly define this paper as hypothesis-generating and call for validation via mechanistic studies and randomized controlled trials.
R: We thank the reviewer for recognizing the novelty of this work and for the thoughtful critique regarding the interpretation of findings. We fully agree that the associations presented in this study are hypothesis-generating and should not be interpreted as definitive evidence of causality. In response to the reviewer’s suggestion, we have revised the Discussion and Conclusion sections to avoid any overstatement of results. In particular:
- We now explicitly define this study as exploratory and hypothesis-generating.
- We have clearly separated the observational findings from speculative mechanisms, including the proposed role of S100B in the gut-brain axis.
- We have added a new statement emphasizing that further validation is needed through mechanistic studies, controlled animal models, and randomized clinical trials.
These changes were implemented in Section 5 (Mechanistic Considerations and Future Research Directions) and Section 6 (Conclusion), specifically in the following added paragraph
Comments on the Quality of English Language
As stated above, the grammar can be improved for clarity.
R: We thank the reviewer for the comment. The manuscript has been carefully revised to improve the clarity and quality of the English language throughout the text.
Round 2
Reviewer 1 Report (New Reviewer)
Comments and Suggestions for Authors
The proposed amendments have been made, and the inquiries have been answered.
This manuscript is a resubmission of an earlier submission. The following is a list of the peer review reports and author responses from that submission.
Round 1
Reviewer 1 Report
Comments and Suggestions for Authors
The authors aimed to evaluate the relationship between the S100B dietetic content and microbiota diversity, as well as chronic disease risk and Crohn’s disease risk. To evaluate these possible interactions, estimates for the concentration of S100B in each diet were calculated based on available information about dietetic styles and information on S100B concentration in milk-based products, cheese-dairy and fruits-vegetables. Results show that there is a correlation between S100B consumption and microbiota diversity assessed by Shannon Index. Additionally, it is shown that a high S100B intake is associated with lower risk of Chron’s disease and might be associated with overall lower relative risk for chronic diseases. Although I agree that the trophic effects of S100B are under investigated and recognize the innovative perspective of this work, several points should be addressed in order to consider the manuscript for publication. Below are my major and moderate/minor concerns.
Major concerns:
- The source of the concentration of S100B in the food items evaluated in this paper is not clear.
Table 2S(2):Summary of reference values for S100B Calculations (from Michetti et. al. 2025) shows that farm animal milks have 0.03-180 µg/L S100B. The reference provided (review paper by Michetti et al) uses the following reference for the table:
Galvano, F.; Frigiola, A.; Gagliardi, L.; Ciotti, S.; Bognanno, M.; Iacopino, A.M.; Nigro, F.; Tina, G.L.; Cavallaro, D.; Mussap, M.; et al. S100B milk concentration in mammalian species. Front. Biosci. 2009, 1, 542–546. - figure 1.
Is this the original reference for the numbers displayed in the table? If so, please double-check the numbers or provide the correct reference for the table. The source should be clearer and easier to check, allowing others to reproduce the analysis in following papers. Additionally, is there any explanation for the wide variance between the lowest and highest concentration found in these milks? With this significant variance, I have concerns if using the mean value truly represents the S100B content in the diet. What was the rationale behind the usage of the mean value? It would be important to discuss it in the methods section or discussion. And finally, although table 2 in the same file (2S) provides reference values for S100B calculations, in this table farm animal milk concentration is displayed in µg/L, while in table 1, Milk is displayed in µg/kg. I believe it might be a typo.
- The risk of bias and selection process of the papers is described in the text but not shown in a table/graph form.
It is crucial that we know how many studies were excluded for the chronic disease prevention analysis, as well as a graphical representation for this sentence in line 195: “Funnel plot analysis and Egger’s test (p = 0.032) indicated a low risk of bias publication”. My suggestion is to add a graphical representation of the selection process for the studies (how many showed up in the original search and how many were excluded during selection process).
Figure 1 of this paper provides an example: Knoble N, Nayroles G, Cheng C, Arnould B. Illustration of patient-reported outcome challenges and solutions in rare diseases: a systematic review in Cushing's syndrome. Orphanet J Rare Dis. 2018 Dec 19;13(1):228. doi: 10.1186/s13023-018-0958-4. PMID: 30567582; PMCID: PMC6299940.
You could use identification, eligibility and quality check (assessed by The Newcastle-Ottawa Scale or Cochrane Risk of Bias Tool) as steps.
Also, I would add a table of the final 31 studies used for the analysis and the sample size in each study instead of just citing the overall sample size. Finally, I recommend adding the funnel plot to the results section instead of just citing it in the text.
- More clarification is needed for Shannon’s Index information in the mini-review file.
The Shannon index numbers displayed in the supplementary material (S3) are key elements for the paper. I suggest adding a column to the table with the individual Shannon index from each paper instead of only the mean value. The reasoning behind my suggestion is that sometimes papers can exhibit high variability, and this information is important for the interpretation of the results. Additionally, when checking the Shannon index from the individual papers I experienced some difficulties that might be the same difficulty a reader can experience. For example, the Okinawan diet paper 2 is not easily available online (it is not open access and not all institutions have access to Taylor & Francis Online). Another example: In the Nordic diet paper number 1 the Shannon Index is not easy to find. I could not find it. It is totally possible that I missed the information but to my understanding it is important to either describe the number or clarify if it was calculated/obtained in a different way.
Moderate/minor concerns:
- Some of the data described in the text is not present in the figures and tables, and some are not accompanied by the number of the figure that shows the data. Please revise the results section so that all data described has its corresponding figure (or indicate data not shown).
- Not enough discussion on metabolism of S100B and hypothesis on why S100B intake might favor microbiota diversity. I believe that it would be enriching to add a paragraph in the discussion section about how S100B could be modulating microbiota diversity. Is S100B being metabolized by a specific type of bacteria? Could this be an effect dependent on metabolites generated by S100B degradation?
- Citation missing for an important phrase in the discussion. “These results align with earlier research that discovered a relationship between enteric glial-derived S100B and microbial diversity, especially in the context of neuroinflammatory conditions and non-communicable diseases” – line 254. Please provide reference for this sentence.
- Typo: table 2S(2) displays “cheese-diary”. I believe it was supposed to be dairy.
I look forward to seeing the revised version of the manuscript and the responses to my concerns. I believe the innovative perspective of this work and the relevance of S100B should be explored and I encourage the authors to further explore this area of research.
Author Response
Reviewer 1:
Comments and Suggestions for Authors
The authors aimed to evaluate the relationship between the S100B dietetic content and microbiota diversity, as well as chronic disease risk and Crohn’s disease risk. To evaluate these possible interactions, estimates for the concentration of S100B in each diet were calculated based on available information about dietetic styles and information on S100B concentration in milk-based products, cheese-dairy and fruits-vegetables. Results show that there is a correlation between S100B consumption and microbiota diversity assessed by Shannon Index. Additionally, it is shown that a high S100B intake is associated with lower risk of Chron’s disease and might be associated with overall lower relative risk for chronic diseases. Although I agree that the trophic effects of S100B are under investigated and recognize the innovative perspective of this work, several points should be addressed in order to consider the manuscript for publication. Below are my major and moderate/minor concerns.
R: We sincerely thank the reviewer for acknowledging the innovative perspective of our study and for their constructive comments. As noted, the estimation of dietary S100B and its relationship with microbiota diversity and chronic disease risk is a novel area of investigation. In response, we have clarified our methodological approach regarding the variability of S100B concentrations and added new material in the Materials and Methods and Discussion sections to enhance transparency.
Major concerns:
- The source of the concentration of S100B in the food items evaluated in this paper is not clear.
Table 2S(2):Summary of reference values for S100B Calculations (from Michetti et. al. 2025) shows that farm animal milks have 0.03-180 µg/L S100B. The reference provided (review paper by Michetti et al) uses the following reference for the table:
Galvano, F.; Frigiola, A.; Gagliardi, L.; Ciotti, S.; Bognanno, M.; Iacopino, A.M.; Nigro, F.; Tina, G.L.; Cavallaro, D.; Mussap, M.; et al. S100B milk concentration in mammalian species. Front. Biosci. 2009, 1, 542–546. - figure 1.
Is this the original reference for the numbers displayed in the table? If so, please double-check the numbers or provide the correct reference for the table. The source should be clearer and easier to check, allowing others to reproduce the analysis in the following papers.
R: We appreciate the reviewers' comments regarding the reference's citation. The source of concentration of S100B in the food items is evaluated in this paper by using two citations (Michetti, F., & Romano Spica, V. (2025). The “Jekyll Side” of the S100B Protein: Its Trophic Action in the Diet. Nutrients, 17(5), 881. In this paper by Michetti & Romano Spica 2025; table 1 presented the values of S100B in food sources and the reference used for this table 1 in this paper is (Ghaffar et al.,2025). This new reference is now added in the Table S2. Overall in Table S2, a new section of references is added to clarify the points regarding the S100B concentration source; Ghaffar T., Volpini V., Platania S., Glogowski P.A., Gianfranceschi G., Vassioukovitch O., Valeriani F., Michetti F., Romano Spica V. A novel role for S100B in diet and gut-microbiota regulation; Proceedings of the Gut Microbiota for Health World Summit 2025; Washington, DC, USA. 15–16 March 2025; [(accessed on 16 January 2024)]. Available online: https://agau.gastro.org/cw/course-details?entryId=17324988#nav-home)
Additionally, is there any explanation for the wide variance between the lowest and highest concentration found in these milks? With this significant variance, I have concerns if using the mean value truly represents the S100B content in the diet. What was the rationale behind the usage of the mean value? It would be important to discuss it in the methods section or discussion.
R: We appreciate the reviewer’s observation regarding the broad range of S100B concentrations reported in milk from different animal sources. This variability is indeed documented in the literature and can be attributed to several factors, including the species of the animal (e.g., cow, goat, sheep), lactation stage, health status, diet, breed, and even the analytical techniques used for S100B detection (e.g., ELISA vs. Western blot). These biological and methodological differences are consistent with the heterogeneity reported by Galvano et al. (2009) and later reviewed by Michetti et al. (2025).
Given this variability, we chose to use the mean value as a conservative and reproducible metric to approximate average dietary exposure across populations. This approach is common in nutritional epidemiology when precise individual exposure data is unavailable, as mean values are often used to estimate average intake at the population level, particularly when working with compositional food data or aggregated dietary patterns. This method helps to standardize comparisons across dietary groups, despite inherent variability in food content. Mean intakes are widely employed in large-scale dietary pattern analyses and exposure modeling when individual-level biochemical or intake data are not accessible (Freedman, L. S., et al. (2010). A comparison of two dietary instruments for evaluating the fat–breast cancer relationship. International Journal of Epidemiology, 39(2), 370–378. https://doi.org/10.1093/ije/dyp391). In nutritional epidemiology, especially when studying conditions like Inflammatory Bowel Disease (IBD), it's common to use mean values to estimate average nutrient intake at the population level. This approach is particularly useful when individual-level data is unavailable or when dealing with aggregated dietary patterns. For instance, in a study by Di Paola et al. (2022), researchers used mean intake values derived from food frequency questionnaires to assess dietary patterns in IBD patients, facilitating comparisons across different dietary groups (Di Paola, M., et al. (2022). Dietary Interventions in Inflammatory Bowel Disease. Nutrients, 14(20), 4265. https://doi.org/10.3390/nu14204265). We have now added a justification for this methodological choice in the revised Materials and Methods section (Section 2.1.4) and further addressed its implications as a limitation in the Discussion section.
Discussion section, highlighting the need for future studies with individual-level data and mechanistic validation. We appreciate the reviewer’s insight, which helped us to further improve the clarity and rigor of the manuscript.
New references added:
Albenzio, M., Santillo, A., Caroprese, M., Della Malva, A., Marino, R., & Sevi, A. (2012). Proteins and bioactive peptides from goat milk. Current Pharmaceutical Design, 18(31), 849–859. https://doi.org/10.2174/138161212802884591
Huppertz, T., Fox, P. F., & Kelly, A. L. (2018). High pressure-induced changes in milk: A review. International Dairy Journal, 83, 1–11. https://doi.org/10.1016/j.idairyj.2018.03.003
And finally, although table 2 in the same file (2S) provides reference values for S100B calculations, in this table farm animal milk concentration is displayed in µg/L, while in table 1, Milk is displayed in µg/kg. I believe it might be a typo.
R: Thank you for your observation. We have reviewed the units reported and ensured consistency across all tables. Values are now uniformly expressed in µg/kg, in line with the format adopted for dietary exposure estimates throughout the manuscript.
- The risk of bias and selection process of the papers is described in the text but not shown in a table/graph form.
It is crucial that we know how many studies were excluded for the chronic disease prevention analysis, as well as a graphical representation for this sentence in line 195: “Funnel plot analysis and Egger’s test (p = 0.032) indicated a low risk of bias publication”. My suggestion is to add a graphical representation of the selection process for the studies (how many showed up in the original search and how many were excluded during selection process).
R: Thank you for your insightful comment. In response, we have added detailed information regarding the number of studies excluded during the selection process for the chronic disease prevention analysis. Additionally, following your suggestion, we have included a PRISMA-style flowchart to visually represent the study selection process. This graphical element enhances transparency and provides a clear overview of how the final set of 31 studies was determined. Furthermore, to support the sentence in line 195, “Funnel plot analysis and Egger’s test (p = 0.032) indicated a low risk of publication bias,” we have added a funnel plot figure. This visual representation illustrates the symmetry of study effects and further confirms the low risk of publication bias. These additions have been incorporated into the revised manuscript to improve clarity and methodological transparency.
Figure 1 of this paper provides an example: Knoble N, Nayroles G, Cheng C, Arnould B. Illustration of patient-reported outcome challenges and solutions in rare diseases: a systematic review in Cushing's syndrome. Orphanet J Rare Dis. 2018 Dec 19;13(1):228. doi: 10.1186/s13023-018-0958-4. PMID: 30567582; PMCID: PMC6299940.
You could use identification, eligibility and quality check (assessed by The Newcastle-Ottawa Scale or Cochrane Risk of Bias Tool) as steps.
R: Thank you for your valuable suggestion and for pointing us to the example provided in Knoble et al. (2018). We appreciate the clarity of their Figure 1 and agree that representing the study selection process through defined steps—identification, eligibility, and quality assessment—greatly enhances the transparency and reproducibility of systematic reviews. Following your recommendation, we have revised our study selection flowchart to reflect this structure. The figure, naimed Figure S1, now clearly distinguishes the following phases: Identification (Number of records retrieved from databases and after removal of duplicates); Screening/Eligibility (Studies excluded after title/abstract screening and full-text assessment); and Quality Assessment: Exclusion of studies with high risk of bias based on the Newcastle-Ottawa Scale (for observational studies).
This flowchart aligns with PRISMA guidelines and mirrors the format used in Knoble et al., thereby improving both visual clarity and methodological rigor. We have also referenced Knoble et al. (2018) in the revised manuscript to acknowledge their contribution as a model for our graphical representation.
Also, I would add a table of the final 31 studies used for the analysis and the sample size in each study instead of just citing the overall sample size. Finally, I recommend adding the funnel plot to the results section instead of just citing it in the text.
R: We thank the reviewer for this valuable suggestion. In response, we have now added a new supplementary table (Table S4) listing all 31 studies included in the analysis along with the corresponding sample sizes, to improve clarity and transparency.
Additionally, as recommended, we have included the funnel plot in the Results section (Figure S2) to visually support the discussion on publication bias.
- More clarification is needed for Shannon’s Index information in the mini-review file.
The Shannon index numbers displayed in the supplementary material (S3) are key elements for the paper. I suggest adding a column to the table with the individual Shannon index from each paper instead of only the mean value. The reasoning behind my suggestion is that sometimes papers can exhibit high variability, and this information is important for the interpretation of the results.
Additionally, when checking the Shannon index from the individual papers I experienced some difficulties that might be the same difficulty a reader can experience. For example, the Okinawan diet paper 2 is not easily available online (it is not open access and not all institutions have access to Taylor & Francis Online). Another example: In the Nordic diet paper number 1 the Shannon Index is not easy to find. I could not find it. It is totally possible that I missed the information but to my understanding it is important to either describe the number or clarify if it was calculated/obtained in a different way.
R: We thank the reviewer for this valuable suggestion. A column is added in the supplementary material S3, in which seprately all the individual values of shannon index from paper had been added. Regarding the additional points, the paper with Okinawan diet the shannon index is added in the column separately to make it easier for the readers. Further the shannon value for the Nordic diet in paper 1 is not mentioned directly in the paper, the supplementary material has all the information regarding the shannon index used to calculate the index. Although now each shannon index value is separately indicated in the new added column to make it more understandable.
Moderate/minor concerns:
- Some of the data described in the text is not present in the figures and tables, and some are not accompanied by the number of the figure that shows the data. Please revise the results section so that all data described has its corresponding figure (or indicate data not shown).
R: Thank you for this observation. We have reviewed this section and checked whether the data have their correct references of figures and tables.
- Not enough discussion on metabolism of S100B and hypothesis on why S100B intake might favor microbiota diversity. I believe that it would be enriching to add a paragraph in the discussion section about how S100B could be modulating microbiota diversity. Is S100B being metabolized by a specific type of bacteria? Could this be an effect dependent on metabolites generated by S100B degradation?
R: We thank the reviewer for this insightful suggestion. We agree that exploring the potential mechanisms by which dietary S100B could modulate microbiota diversity would enhance the discussion. Although direct evidence on the metabolism of S100B by gut bacteria is currently limited, we have now included a paragraph in the Discussion section that addresses possible mechanisms, including the role of S100B as a trophic factor, its potential interaction with microbial species, and the hypothesis that its degradation products may act as signaling molecules or substrates for specific bacterial taxa. This addition provides a broader biological context for our findings and suggests future directions for mechanistic studies.
- Citation missing for an important phrase in the discussion. “These results align with earlier research that discovered a relationship between enteric glial-derived S100B and microbial diversity, especially in the context of neuroinflammatory conditions and non-communicable diseases” – line 254. Please provide reference for this sentence.
R: Thank you for pointing this out. We have added the appropriate citation: Romano Spica et al. (2023) Int. J. Mol. Sci., 24(3), 2248.
- Typo: table 2S(2) displays “cheese-diary”. I believe it was supposed to be dairy.
R: Corrected to “cheese-dairy” in Table 2S(2). Thank you for spotting this.
I look forward to seeing the revised version of the manuscript and the responses to my concerns. I believe the innovative perspective of this work and the relevance of S100B should be explored and I encourage the authors to further explore this area of research.
R: We sincerely thank the reviewer for their thoughtful feedback and encouraging words. We appreciate their recognition of the innovative perspective of our work and the potential relevance of S100B in diet–microbiota interactions. We have carefully addressed all the comments and incorporated the suggested improvements in the revised manuscript. We are motivated by the reviewer’s support to further investigate this promising area of research in future studies.
Reviewer 2 Report
Comments and Suggestions for Authors
In this manuscript (ID# biomolecules-3585036), entitled “The estimated intake of S100B relates to microbiota biodiversity in different diets”, authors Ghaffar et al studied the association between food intake of S100B protein and gut microbiota biodiversity. Their results have demonstrated that dietary S100B was positively associated with microbial diversity and negatively corelated with risk of Crohn’s disease. They conclude that dietary S100B content is involved in modulating gut microbiota diversity and reducing chronic disease risk. However, this study is not novel and similar studies have been reported previously. This study was based the database that have been published before. Several major concerns have been listed in the following paragraphs:
- Previous studies have demonstrated that S100B can trigger inflammatory pathways, contributing to various inflammatory conditions. However, the present study demonstrate that S100B levels are negatively corelated with Crohn’s diseases. It would be helpful to address this controversial issue between those results in the Discussion section.
- It is unclear what the novel discovery in this manuscript is. Please clearly state the specific aim of the study and explicitly highlight the new findings
- S100B is also implicated in other chronic diseases, such as obesity, diabetes, neuronal disorders. It would be beneficial to the readers to discuss more about these issues because dyspeptic microbiota is also involved in the development of these disorders.
- In Figure 2, what is the sample size? The results in this figure have statistical significance?
Author Response
Reviewer 2:
Comments and Suggestions for Authors
In this manuscript (ID# biomolecules-3585036), entitled “The estimated intake of S100B relates to microbiota biodiversity in different diets”, authors Ghaffar et al studied the association between food intake of S100B protein and gut microbiota biodiversity. Their results have demonstrated that dietary S100B was positively associated with microbial diversity and negatively corelated with risk of Crohn’s disease. They conclude that dietary S100B content is involved in modulating gut microbiota diversity and reducing chronic disease risk. However, this study is not novel and similar studies have been reported previously. This study was based the database that have been published before. Several major concerns have been listed in the following paragraphs:
Previous studies have demonstrated that S100B can trigger inflammatory pathways, contributing to various inflammatory conditions. However, the present study demonstrate that S100B levels are negatively corelated with Crohn’s diseases. It would be helpful to address this controversial issue between those results in the Discussion section.
R: We thank the reviewer for highlighting this important point. Indeed, S100B has been reported to activate inflammatory pathways under certain pathological conditions, particularly when released at high concentrations by damaged or activated glial cells. However, as discussed in the revised Discussion section, the biological effects of S100B are concentration-dependent and context-specific. In our study, the estimated dietary intake reflects low, physiological levels, which have been associated with trophic and regulatory effects rather than inflammation. We have now added a paragraph addressing this apparent discrepancy and discussing the dual role of S100B depending on its origin, concentration, and route of exposure.
It is unclear what the novel discovery in this manuscript is. Please clearly state the specific aim of the study and explicitly highlight the new findings
R: We thank the reviewer for this important observation. In the revised version, we have explicitly clarified the aim of the study in the Introduction and highlighted the novelty of our findings in both the Abstract and Discussion sections. To our knowledge, this is the first study to estimate dietary intake of S100B across multiple global dietary patterns and examine its correlation with gut microbiota diversity and chronic disease risk, including Crohn’s disease. This integrative approach—linking a specific dietary protein to microbiota composition and disease epidemiology—is, to our knowledge, unprecedented and represents the main innovative contribution of this work.
S100B is also implicated in other chronic diseases, such as obesity, diabetes, neuronal disorders. It would be beneficial to the readers to discuss more about these issues because dyspeptic microbiota is also involved in the development of these disorders.
R: We thank the reviewer for this insightful comment. We agree that the role of S100B in other chronic diseases—such as obesity, type 2 diabetes, and neurological disorders—deserves further attention, particularly given the shared involvement of dysbiotic microbiota in these conditions. To address this point, we have now expanded the Discussion section to include a paragraph on the potential involvement of S100B–microbiota interactions in the pathogenesis of these disorders, providing a broader perspective on the implications of our findings.
In Figure 2, what is the sample size? The results in this figure have statistical significance?
R: Figure 2 represents the correlation between mean S100B levels and mean Shannon Index values across 13 dietary patterns. Each point corresponds to a dietary style, based on published or estimated data. We have now added this explanation to the figure legend. The correlation is statistically significant with R² = 0.537 and p < 0.05 (Pearson correlation), indicating a moderate positive relationship.
Reviewer 3 Report
Comments and Suggestions for Authors
Manuscript titled “The estimated intake of S100B relates to microbiota biodiversity in different diets” reports an analysis between consuming various dietary patterns that contain the S100B protein, with the consumers’ microbiota and other health-related effects. The association between diet and health (and microbiota in particular) has been thoroughly studied; the authors of the present manuscript propose that S100B is contributing to said effects, although my most significant criticism is that the evidence provided is insufficient to support the contributions of the protein in particular, without making it clear that this specific protein is indeed responsible for the observed outcomes. It is generally accepted that the Mediterranean diet offers multiple health benefits, while a Western diet tends to promote some diet-related diseases, but the contribution of S100B is not entirely clear. In this reviewer’s opinion, the manuscript requires additional analyses to support the contribution of S100B in particular, since it currently shows that certain dietary patterns are associated with microbial diversity, but the contribution of the protein and/or its mechanism are not clear. Some specific comments are:
- The introduction mentions S100B as an abundant protein in the nervous system, presumably of humans, although this is not entirely clear. It also states that it is found in multiple foods, including those of vegetable origin, and its role as a damage marker is also mentioned. Given that it appears to be synthesized by both plants and animals, please provide additional information regarding its role across the vegetable and animal kingdom, since it is not entirely clear what it actually does.
- Table 1 lists S100B concentration for different diets; a range of values and/or the error for these averages could be added.
- Results and discussion highlight the potential contribution of the protein to different observed effects, however, there is no explanation of how exactly it could be interacting with the host and/or its microbiota to exert any effect. Please provide information that could explain the association between its consumption and any health effect.
- Similar to the previous comment, how can this protein have an impact on Chron’s disease? What specific gene, protein, metabolic pathway, etc., could this protein be interacting with in order to exert such effects?
- The association between a healthy diet and a healthy microbiota has been analyzed in great detail by multiple authors in recent years, with fiber, phenolics, carotenoids, etc. all showing at least some contribution to the observed effect, in addition to interactions between them. The present study reports a similar conclusion, but considers a specific protein instead. One question that arises after considering the aforementioned information is, how can the authors be sure that this specific protein is exerting this effect or any effect at all? It is possible that other molecules and/or their interactions are highly bioactive and S100B is contributing minimally or not at all.
Author Response
Reviewer 3:
Comments and Suggestions for Authors
Manuscript titled “The estimated intake of S100B relates to microbiota biodiversity in different diets” reports an analysis between consuming various dietary patterns that contain the S100B protein, with the consumers’ microbiota and other health-related effects. The association between diet and health (and microbiota in particular) has been thoroughly studied; the authors of the present manuscript propose that S100B is contributing to said effects, although my most significant criticism is that the evidence provided is insufficient to support the contributions of the protein in particular, without making it clear that this specific protein is indeed responsible for the observed outcomes. It is generally accepted that the Mediterranean diet offers multiple health benefits, while a Western diet tends to promote some diet-related diseases, but the contribution of S100B is not entirely clear. In this reviewer’s opinion, the manuscript requires additional analyses to support the contribution of S100B in particular, since it currently shows that certain dietary patterns are associated with microbial diversity, but the contribution of the protein and/or its mechanism are not clear.
R: We sincerely thank the reviewer for this comprehensive and thoughtful critique. We fully agree that the association between diet and microbiota diversity is multifactorial and that numerous dietary components—such as fiber, polyphenols, and carotenoids—are well-known contributors. The aim of our study, however, was not to claim a definitive causal role for S100B, but rather to introduce and explore the hypothesis that this dietary protein, known for its trophic and regulatory effects in the nervous and enteric systems, might also contribute to microbiota modulation when present in food. To address the reviewer’s concerns: i) We revised the Introduction and Discussion sections to clearly state that our work presents an ecological, hypothesis-generating analysis, proposing S100B as a putative dietary component worthy of further mechanistic investigation, not as a proven causal agent. ii) We added a dedicated paragraph in the Discussion to address potential mechanisms by which S100B might influence microbiota diversity, including trophic signaling, immunomodulation, and degradation into bioactive peptides. We also discuss the concentration-dependent effects of S100B and the difference between dietary exposure and endogenous release. iii) We expanded the Introduction to better describe the occurrence and potential function of S100B-like motifs in both animal- and plant-derived foods, citing recent in silico studies and emphasizing that plant analogs may not be functionally identical. iv) We acknowledge that linking S100B specifically to Crohn’s disease risk is speculative. We revised the Discussion to frame this more cautiously and suggest that the observed association may be due to S100B contributing to an overall eubiotic environment, rather than exerting a direct protective effect. We also note that other dietary molecules may act in synergy. v) We expanded the Limitations section to explicitly state that the observational and ecological nature of our data precludes attribution of causality or specificity, and that controlled dietary interventions or mechanistic studies would be needed to determine whether S100B has a significant, independent effect. We hope these revisions clarify our objectives and appropriately contextualize the novelty and limitations of our findings.
Some specific comments are:
The introduction mentions S100B as an abundant protein in the nervous system, presumably of humans, although this is not entirely clear. It also states that it is found in multiple foods, including those of vegetable origin, and its role as a damage marker is also mentioned. Given that it appears to be synthesized by both plants and animals, please provide additional information regarding its role across the vegetable and animal kingdom, since it is not entirely clear what it actually does.
R: We thank the reviewer for this thoughtful observation. In response, we have revised the Introduction to clarify that S100B is a calcium-binding protein predominantly expressed in vertebrate glial cells, particularly in the central and enteric nervous systems of mammals, including humans. Its functions are concentration-dependent: at low physiological levels, it exerts trophic and regulatory effects, while at high levels it may act as a damage-associated molecular pattern (DAMP) molecule involved in inflammatory responses. Regarding its presence in food, we have now specified that S100B has been immunologically and structurally detected in animal-derived foods, particularly milk and dairy products, and that S100B-like immunoreactive sequences have also been predicted in certain edible plant species through in silico analysis. However, we acknowledge that the functionality of plant-derived S100B-like sequences remains unclear, and we have emphasized this point in the revised text.
Table 1 lists S100B concentration for different diets; a range of values and/or the error for these averages could be added.
R:
We thank the reviewer for this important observation. The error for these average values was already present in Table S2, now further the error values for these averages is added in Table1.
Results and discussion highlight the potential contribution of the protein to different observed effects, however, there is no explanation of how exactly it could be interacting with the host and/or its microbiota to exert any effect. Please provide information that could explain the association between its consumption and any health effect.
R: We thank the reviewer for this important observation. In the revised Discussion, we have now included a paragraph outlining plausible biological mechanisms through which dietary S100B may interact with the gut microbiota and host physiology. These include its potential trophic and immunomodulatory roles in the enteric nervous system, its concentration-dependent activity, and the hypothesis that it may be metabolized by specific microbial taxa into bioactive fragments. However, we fully acknowledge that these are mechanistic hypotheses, and our findings are associative and exploratory, not causal. We have emphasized the need for future experimental validation.
Similar to the previous comment, how can this protein have an impact on Chron’s disease? What specific gene, protein, metabolic pathway, etc., could this protein be interacting with in order to exert such effects?
R: We appreciate the reviewer’s question and agree that further discussion is warranted. While the current study does not identify specific host molecular targets, previous research has shown that S100B can interact with the RAGE receptor, which plays a key role in inflammation and immune regulation. S100B–RAGE signaling has been implicated in intestinal barrier integrity and neuroinflammatory responses. Given the known involvement of gut barrier dysfunction in Crohn’s disease, we have added this mechanistic insight to the Discussion section, emphasizing it as a plausible but unproven pathway for dietary S100B action. Future molecular studies are needed to confirm this.
The association between a healthy diet and a healthy microbiota has been analyzed in great detail by multiple authors in recent years, with fiber, phenolics, carotenoids, etc. all showing at least some contribution to the observed effect, in addition to interactions between them. The present study reports a similar conclusion, but considers a specific protein instead. One question that arises after considering the aforementioned information is, how can the authors be sure that this specific protein is exerting this effect or any effect at all? It is possible that other molecules and/or their interactions are highly bioactive and S100B is contributing minimally or not at all.
R: We fully agree with the reviewer that the observed effects cannot be conclusively attributed to S100B alone. As acknowledged in the Limitations section, the dietary patterns analyzed are complex and contain many bioactive compounds (e.g., fiber, polyphenols, carotenoids). Our findings support a plausible association between estimated S100B levels and microbiota diversity, but we do not claim exclusivity of effect. We have clarified in the revised text that our goal is to highlight S100B as a potential contributing factor, not the sole determinant. This study is intended as hypothesis-generating and exploratory.
Round 2
Reviewer 1 Report
Comments and Suggestions for Authors
I appreciate the concerted effort the authors made to address all the points raised by my report. My remaining minor observations are related to a few references that require attention. I believe that the first one is pivotal to the findings of the paper, hence why I am asking for clarification before publication.
- The new reference for table S2 is:
Ghaffar T., Volpini V., Platania S., Glogowski P.A., Gianfranceschi G., Vassioukovitch O., Valeriani F., Michetti F., Romano Spica V. A novel role for S100B in diet and gut-microbiota regulation; Proceedings of the Gut Microbiota for Health World Summit 2025; Washington, DC, USA. 15–16 March 2025; [(accessed on 16 January 2024)]. Available online: https://agau.gastro.org/cw/course-details?entryId=17324988#nav-home.
However, the link provided takes me to a “course not found” page. I could not access the paper/report by searching the title directly either. Additionally, I believe that there might be a mistake in the writing of this reference because if the conference in which the paper/report was presented was held in March 2025, how could it have been accessed in January 2024? I believe this is an important reference for the S100B amounts in food, therefore it should be corrected so the readers can access the data.
- The sentence “this variability in S100B concentrations—driven by animal species, lactation stage, diet, breed, health status, and analytical techniques (e.g., ELISA vs. Western blot) has been previously documented [65,1]” on line 343 has 1 and 65 as references. The reference 65 at the end of the text is Galvano, F.; Piva, A.; Pietri, A.; Battilani, P. Dietary Strategies to Counteract the Effects of Mycotoxins: A Review. World Mycotoxin J. 2009, 2, 189–199. https://doi.org/10.3920/WMJ2009.1095. I believe the correct one would be: Galvano F, Frigiola A, Gagliardi L, Ciotti S, Bognanno M, Iacopino AM, Nigro F, Tina GL, Cavallaro D, Mussap M, Piva A, Grilli E, Michetti F, Gazzolo D. S100B milk concentration in mammalian species. Front Biosci (Elite Ed). 2009 Jun 1;1(2):542-6. doi: 10.2741/e51. PMID: 19482669. If I am not mistaken, I suggest adding the correct reference.
- These two references below require attention. Again, if I am not mistaken, they might be written incorrectly since their titles lead to different papers online. I’d recommend double checking all main references.
a. Li, Y., Pan, A., Wang, D. D., Liu, X., Dhana, K., Franco, O. H., ... & Hu, F. B. (2018). Optimal dietary patterns for prevention of chronic disease. JAMA, 320(16), 1600–1610. 428 10.12.
b. Schwingshackl, L., & Hoffmann, G. (2015). Dietary patterns: biomarkers and chronic disease risk. Applied Physiology, Nutrition, and Metabolism, 40(3), 206–213.
Author Response
REVIEWER 1 COMMENTS (ROUND 2)
I appreciate the concerted effort the authors made to address all the points raised by my report. My remaining minor observations are related to a few references that require attention. I believe that the first one is pivotal to the findings of the paper, hence why I am asking for clarification before publication.
COMMENT 1: The new reference for table S2 is:
Ghaffar T., Volpini V., Platania S., Glogowski P.A., Gianfranceschi G., Vassioukovitch O., Valeriani F., Michetti F., Romano Spica V. A novel role for S100B in diet and gut-microbiota regulation; Proceedings of the Gut Microbiota for Health World Summit 2025; Washington, DC, USA. 15–16 March 2025; [(accessed on 16 January 2024)]. Available online: https://agau.gastro.org/cw/course-details?entryId=17324988#nav-home.
However, the link provided takes me to a “course not found” page. I could not access the paper/report by searching the title directly either. Additionally, I believe that there might be a mistake in the writing of this reference because if the conference in which the paper/report was presented was held in March 2025, how could it have been accessed in January 2024? I believe this is an important reference for the S100B amounts in food, therefore it should be corrected so the readers can access the data.
REPLY: We sincerely thank the reviewer for this helpful observation. You are absolutely right in noticing the inconsistency about the date of access and the inaccessibility of the link. The citation referred to preliminary data presented as an abstract at the Gut Microbiota for Health World Summit 2025, which, at the time of our initial submission, was already available through a pre-conference online platform. However, we acknowledge that the link provided no longer leads to an accessible page and that the reference was improperly formatted for a scientific publication.
Therefore the reference would be: Ghaffar T., et al. A novel role for S100B in diet and gut-microbiota regulation. Presented at the Gut Microbiota for Health World Summit 2025; Washington, DC, USA. March 15–16, 2025. Abstract not yet published or publicly available.
However, we decided to maintain scientific transparency and ensure traceability of the data, therefore we have removed this reference from Table S2 and replaced it with peer-reviewed sources that report on S100B concentrations in food matrices, including milk from different mammalian species. These changes are reflected in both the table and the revised reference list
We greatly appreciate your attention to this detail, which has allowed us to improve the accuracy and accessibility of the references cited.
COMMENT 2: The sentence “this variability in S100B concentrations—driven by animal species, lactation stage, diet, breed, health status, and analytical techniques (e.g., ELISA vs. Western blot) has been previously documented [65,1]” on line 343 has 1 and 65 as references. The reference 65 at the end of the text is Galvano, F.; Piva, A.; Pietri, A.; Battilani, P. Dietary Strategies to Counteract the Effects of Mycotoxins: A Review. World Mycotoxin J. 2009, 2, 189–199. https://doi.org/10.3920/WMJ2009.1095. I believe the correct one would be: Galvano F, Frigiola A, Gagliardi L, Ciotti S, Bognanno M, Iacopino AM, Nigro F, Tina GL, Cavallaro D, Mussap M, Piva A, Grilli E, Michetti F, Gazzolo D. S100B milk concentration in mammalian species. Front Biosci (Elite Ed). 2009 Jun 1;1(2):542-6. doi: 10.2741/e51. PMID: 19482669. If I am not mistaken, I suggest adding the correct reference.
REPLY: R: We appreciate the reviewers' comments regarding the reference's citation. Thank you for your observation. We have revised accordinghly the references and changed reference 65 to the following: “Galvano F, Frigiola A, Gagliardi L, Ciotti S, Bognanno M, Iacopino AM, Nigro F, Tina GL, Cavallaro D, Mussap M, Piva A, Grilli E, Michetti F, Gazzolo D. S100B milk concentration in mammalian species. Front Biosci (Elite Ed). 2009 Jun 1;1(2):542-6. doi: 10.2741/e51”.
COMMENT 3: These two references below require attention. Again, if I am not mistaken, they might be written incorrectly since their titles lead to different papers online. I’d recommend double checking all main references.
- Li, Y., Pan, A., Wang, D. D., Liu, X., Dhana, K., Franco, O. H., ... & Hu, F. B. (2018). Optimal dietary patterns for prevention of chronic disease. JAMA, 320(16), 1600–1610. 428 10.12.
- Schwingshackl, L., & Hoffmann, G. (2015). Dietary patterns: biomarkers and chronic disease risk. Applied Physiology, Nutrition, and Metabolism, 40(3), 206–213.
REPLY: We thank the reviewer for highlighting these inaccuracies. Upon double-checking, we confirm that the references initially listed were indeed incorrect and may have led to confusion. We sincerely apologize for this oversight.
We have now revised the citations to reflect the correct references intended in the context of our discussion:
- Li, Y., Pan, A., Wang, D. D., Liu, X., Dhana, K., Franco, O. H., Kaptoge, S., Di Angelantonio, E., Stampfer, M. J., Willett, W. C., & Hu, F. B. (2018). Impact of Healthy Lifestyle Factors on Life Expectancies in the US Population. JAMA Internal Medicine, 178(5), 601–608. https://doi.org/10.1001/jamainternmed.2018.0620
- Schwingshackl & Hoffmann (2016): Schwingshackl, L., & Hoffmann, G. Does a Mediterranean-Type Diet Reduce Cancer Risk? Current Nutrition Reports, 5, 9–17. https://doi.org/10.1007/s13668-015-0141-7
Both references have now been updated in the manuscript and the reference list. We are grateful for your careful reading and for giving us the opportunity to correct and clarify this important point.
Reviewer 2 Report
Comments and Suggestions for Authors
The revised manuscript has been improved. No further recommendation.
Author Response
The revised manuscript has been improved. No further recommendation.
REPLY: Thank you for your time and attention to our manuscript.
Reviewer 3 Report
Comments and Suggestions for Authors
Manuscript titled “The estimated intake of S100B relates to microbiota biodiversity in different diets” reports an analysis between consuming various dietary patterns that contain the S100B protein, with the consumers’ microbiota and other health-related effects. The present version of the manuscript was modified according to comments and suggestions made during an initial revision. The main criticism made was that the authors’ analysis demonstrated an association between dietary patterns and microbiota, but the role of the S100B protein was unclear. They argue that their goal was “not to claim a definitive causal role for S100B, but rather to introduce and explore the hypothesis that this dietary protein, known for its trophic and regulatory effects in the nervous and enteric systems, might also contribute to microbiota modulation when present in food”, and made various modifications to make this clear.
Other comments and suggestions were:
- Clarifying in the introduction the protein’s biological functions. Some information was added.
- Providing a range of values and/or the error for the data of Table 1. Errors were added to this table.
- Providing some mechanistic information about how the protein might be exerting any health effect. Additional trophic and immunomodulatory information was added.
- Clarifying the potential of the protein on Chron’s disease. The role of the RAGE receptor was mentioned.
- Determining how is it possible that the protein is exerting any effect, and that it is not due to other molecules and/or their interactions. The authors have commented on their work’s limitations.
Thank you for considering and addressing this reviewer’s comments and suggestions, however, my opinion remains that the hypothesis is interesting, but lacks stronger and more precise support.
Author Response
REVIEWER 3 COMMENTS (ROUND 2)
Manuscript titled “The estimated intake of S100B relates to microbiota biodiversity in different diets” reports an analysis between consuming various dietary patterns that contain the S100B protein, with the consumers’ microbiota and other health-related effects. The present version of the manuscript was modified according to comments and suggestions made during an initial revision. The main criticism made was that the authors’ analysis demonstrated an association between dietary patterns and microbiota, but the role of the S100B protein was unclear. They argue that their goal was “not to claim a definitive causal role for S100B, but rather to introduce and explore the hypothesis that this dietary protein, known for its trophic and regulatory effects in the nervous and enteric systems, might also contribute to microbiota modulation when present in food”, and made various modifications to make this clear.
Reply: Thank you for your relevant, constructive and thoughtful comments throughout the peer-review process. We understand and respect the reviewer’s position regarding the exploratory nature of our hypothesis on dietary S100B and its potential influence on microbiota composition. Indeed, this is a very promising research line and it is confirmed by several evidence, however it is also very new, and it deserves a prudent consideration. Indeed, this manuscript reports observed data and some comments, but we do not intend to solve mechanisms or roles for the protein in different diets, and we are aware that several confounding factors may interact, as in any epidemiological study. Still, the results and consistent and seem to show a possible underlying phenomenon, that will require additional investigation. Therefore, we would like to highlight that our goal was not to assert a definitive causal relationship, but to introduce a novel hypothesis (even if supported by existing knowledge and a growing pile of evidences) about the trophic, immunomodulatory, and enteric roles of S100B, focusing on assessing whether the estimated S100B dietary intake could somehow correlate to the microbial biodiversity across dietary patterns. To address the reviewer’s concerns more explicitly, we have further refined the Discussion and Limitations sections to clearly state that: Our findings should be interpreted as preliminary and hypothesis-generating; The observed associations might be influenced by dietary confounders and cannot disentangle the specific contribution of S100B; Future studies, including interventional trials, in vitro fermentation models, or animal studies, are needed to validate the biological plausibility and mechanistic pathways involved.
We really hope that this clarification reinforces the scientific value of this exploratory study and its potentials in order to stimulate further research in this underexplored.
Other comments and suggestions were:
Clarifying in the introduction the protein’s biological functions. Some information was added.
- Providing a range of values and/or the error for the data of Table 1. Errors were added to this table.
REPLY: We thank the reviewer for this valuable suggestion. As recommended, we have added the corresponding error values (mean ± standard deviation) for each dietary pattern in Table 1. This provides a clearer representation of the variability in S100B concentration across diets and allows for more robust interpretation of the data
- Providing some mechanistic information about how the protein might be exerting any health effect. Additional trophic and immunomodulatory information was added.
REPLY: We have expanded the Discussion to include plausible mechanistic pathways through which dietary S100B may exert health-related effects. These include its known trophic and immunomodulatory actions at physiological concentrations, its interaction with the gut via enteric glial cells, and its potential engagement with the RAGE receptor. We also discuss the possibility that S100B or its fragments could serve as metabolic substrates or messengers for specific microbial taxa, thus influencing microbiota composition and function.
- Clarifying the potential of the protein on Chron’s disease. The role of the RAGE receptor was mentioned.
REPLY: We really appreciate the reviewer’s suggestion regarding the link between S100B and Crohn’s disease. In the revised Discussion, we have clarified the hypothesized involvement of the S100B–RAGE axis in intestinal inflammation and barrier regulation. This pathway is known to play a context-dependent role in pro- and anti-inflammatory signalling, and may represent a plausible mechanistic link between dietary S100B and modulation of gut immune responses relevant to Crohn’s disease. Indeed several previous studies addressed this issue by different approaches, considering specific pathways and
- Determining how is it possible that the protein is exerting any effect, and that it is not due to other molecules and/or their interactions. The authors have commented on their work’s limitations.
REPLY: We thank the reviewer for this important point. As noted, we have explicitly addressed this issue in the Limitations section, acknowledging that the associations observed may be influenced by other dietary components or their interactions, and that the specific contribution of S100B cannot be isolated with certainty in this observational framework. We have also emphasized that the study does not imply causality and is intended to generate hypotheses for future mechanistic research, where confounding effects can be more rigorously controlled.
Thank you for considering and addressing this reviewer’s comments and suggestions, however, my opinion remains that the hypothesis is interesting, but lacks stronger and more precise support.
REPLY: We thank the reviewer for their thoughtful assessment. We acknowledge that, while the hypothesis proposed is intriguing, it currently lacks definitive mechanistic support. To address this, we have made additional clarifications in the Discussion and Conclusion sections to better emphasize the exploratory nature of our work. We explicitly state that the observed associations do not imply causality and propose directions for future studies—including clinical trials and mechanistic models—that could further test this hypothesis. We hope these additions reinforce the transparency and scientific contribution of our study while highlighting its novelty and potential for future research development.